# Effect of high-dose *N*-acetylcysteine on exacerbations and lung function in patients with mild-to-moderate COPD: a double-blind, parallel group, multicentre randomised clinical trial

Evidence for the treatment of patients with mild-to-moderate chronic obstructive pulmonary disease (COPD) is limited. The efficacy of N-acetylcysteine (an antioxidant and mucolytic agent) for patients with mild-to-moderate COPD is uncertain. In this multicentre, randomised, double-blind, placebo-controlled trial, we randomly assigned 968 patients with mild-to-moderate COPD to treatment with N-acetylcysteine (600 mg, twice daily) or matched placebo for two years. Eligible participants were 40-80 years of age and had mild-to-moderate COPD (forced expiratory volume in 1 second [$FEV_1$] to forced vital capacity ratio <0.70 and an $FEV_1 \geq 50\%$ predicted value after bronchodilator use). The coprimary outcomes were the annual rate of total exacerbations and the between-group difference in the change from baseline to 24 months in $FEV_1$ before bronchodilator use. COPD exacerbation was defined as the appearance or worsening of at least two major symptoms (cough, expectoration, purulent sputum, wheezing, or dyspnoea) persisting for at least 48 hours. Assessment of exacerbations was conducted every three months, and lung function was performed annually after enrolment. The difference between the N-acetylcysteine group and the placebo group in the annual rate of total exacerbation were not significant (0.65 vs. 0.72 per patient-year; relative risk [RR], 0.90; 95% confidence interval [CI], 0.80–1.02; $P = 0.10$). There was no significant difference in $FEV_1$ before bronchodilator use at 24 months. Long-term treatment with high-dose N-acetylcysteine neither significantly reduced the annual rate of total exacerbations nor improved lung function in patients with mild-to-moderate COPD. Chinese Clinical Trial Registration: ChiCTR-IIR-17012604.

Chronic obstructive pulmonary disease (COPD) is the third leading cause of death worldwide and thus is a global public health concern[1,2]. In China, the prevalence of spirometry-defined COPD in the population aged 40 years or older increased from 8.2% in 2007 to 13.7% in 2018, with 92% Global Initiative for Chronic Obstructive Lung Disease (GOLD) stage 1 (mild) and 2 (moderate) patients[3,4]. Patients with mild-to-moderate COPD experienced a more rapid decline in lung function than patients with advanced COPD[5–9]. Moreover, moderate-to-severe

e-mail: pxran@gzhmu.edu.cn

exacerbation in mild-to-moderate COPD has a more serious impact on lung function decline than moderate-to-severe exacerbation in advanced COPD[8]. It is urgent to decrease exacerbation risk and lung function decline in patients with mild-to-moderate COPD. However, up to now, evidence for the treatment of patients with mild-to-moderate COPD is limited. There was only one study suggested that tiotropium effectively improved lung function, reduced the rate of decline in lung function, and reduced the incidence of exacerbation in patients with mild-to-moderate COPD[9].

N-acetylcysteine, a mucolytic agent with antioxidant properties[10–13], has previously been found to reduce the incidence of exacerbations in COPD patients without inhaled corticosteroids or in moderate-to-severe COPD patients with a history of frequent exacerbations[14,15] and is well tolerated[14–17]. The efficacy of N-acetylcysteine for patients with mild-to-moderate COPD is still unclear. A dose-response relationship has been observed with N-acetylcysteine in advanced COPD[17]. Therefore, we hypothesized that a higher dose of N-acetylcysteine in patients with mild-to-moderate COPD might lead to a greater improvement in respiratory health outcomes[14,15], and then conducted a randomized controlled trial to investigate the efficacy and safety of long-term treatment with high-dose N-acetylcysteine (600 mg, twice daily) in patients with COPD of GOLD stage 1–2, with the annual rate of exacerbations and lung function as the coprimary outcomes. Here, we report the results of this randomized clinical trial.

## Results

### Characteristics of the patients

This clinical trial has been completed. Between 7 September 2017 and 3 January 2022, we randomly assigned 968 patients with mild-to-moderate COPD to receive high-dose N-acetylcysteine or placebo. Of 968 patients randomized, 924 patients who completed at least once followed-up visit were included in the full analysis set (FAS) for exacerbation (placebo group: $n = 460$; N-acetylcysteine group: $n = 464$), with the mean treatment duration of 608 days in the placebo group and 616 days in the N-acetylcysteine group ($P = 0.56$), and 744 patients who completed at least one followed-up spirometry were included in the FAS for lung function (placebo group: $n = 363$; N-acetylcysteine group: $n = 381$) (Fig. 1). At the end, 656 patients completed follow-up for two years (placebo group: $n = 323$; N-acetylcysteine group: $n = 333$), with a total of 312 patients dropping out of this trial (placebo group: 161 patients; N-acetylcysteine group: 151 patients; $P = 0.49$). There was no significant difference in withdrawal reasons and time to withdraw (Fig. 1 and Supplementary Fig. 1).

Baseline characteristics were comparable between the two groups of patients both in randomization and in the FAS for exacerbation as well as for lung function (Table 1 and Supplementary Table 1, 2). In the FAS for exacerbation, mean age of patients was $62.5 \pm 8.4$ years and $62.6 \pm 8.0$ years, mean forced expiratory volume in 1 s ($FEV_1$) after bronchodilator use was $2.08 \pm 0.54$ L (82.4 % of the predicted $FEV_1$) and $2.04 \pm 0.54$ L (82.0% of the predicted $FEV_1$) in the N-acetylcysteine and placebo groups, respectively (Table 1). Baseline characteristics of the patients who withdrew were comparable in the two groups, except for COPD exacerbations in the previous year (Supplementary Table 3). Among the patients who withdrew from this study, there were fewer incidences of exacerbations in the N-acetylcysteine group than the placebo group (15.2% vs. 24.8%, $P = 0.035$). There was also no difference in medication compliance, smoking status, and respiratory medications for COPD between the two groups at baseline or during the study (Table 1 and Supplementary Table 4). The peripheral blood N-acetylcysteine concentration in the N-acetylcysteine group was significantly higher than in the placebo group (Supplementary Table 5).

### Primary outcomes

The incidence of exacerbations was lower in the N-acetylcysteine group than in the placebo group, but this difference did not reach statistical significance (0.65 [95% CI, 0.56–0.76] vs. 0.72 [95% CI, 0.64–0.84] per patient-year; relative risk [RR], 0.90; 95% CI, 0.80–1.02; $P = 0.10$) (Fig. 2). There was no significant difference in $FEV_1$ before bronchodilator use at 24 months between the N-acetylcysteine group and the placebo group (137 ml vs. 121 ml; difference, 16 ml; 95% CI, −69 to 100; $P = 0.72$) (Fig. 3).

### Secondary outcomes

The annual rate of moderate or severe exacerbations was significantly 24% lower in the N-acetylcysteine group than in the placebo group (0.34 vs. 0.45 per patient-year; RR, 0.76; 95% CI, 0.64–0.90; $P = 0.001$) (Fig. 2). The annual rate of severe exacerbations was similar between the N-acetylcysteine and placebo groups. There was no significant between-group difference in the time to the first exacerbation of COPD (hazard ratio [HR], 0.98; 95% CI, 0.81–1.18; $P = 0.84$), time to the first moderate or severe exacerbation of COPD, and time to first severe exacerbation of COPD (Supplementary Fig. 2).

There was no significant difference in $FEV_1$ before bronchodilator use at 12 months between the N-acetylcysteine group and the placebo group, with a between-group difference of 28 ml (95% CI, −51 to 108; $P = 0.48$). There was no significant difference in $FEV_1$ after bronchodilator use or in forced vital capacity (FVC) before or after bronchodilator use between the N-acetylcysteine group and the placebo group ($P > 0.05$ for all comparisons of $FEV_1$ and FVC) (Fig. 3). No clinically significant differences were observed in the annual decline in $FEV_1$, FVC, $FEV_1/FVC$, $FEV_1$ % of predicted value, and FVC % of predicted value before and after bronchodilator use between the two groups (Table 2).

The COPD Assessment Test (CAT) score and modified Medical Research Council (mMRC) score declined in both groups from the start of the study. However, there were no significant differences between the N-acetylcysteine group and the placebo group in the mMRC score and CAT score at each follow-up time point (Supplementary Table 6, 7).

One hundred seventeen adverse events occurred in 464 patients (25.2%) treated with at least one dose of N-acetylcysteine, and 98 adverse events occurred in 464 patients (21.1%) treated with at least one dose of placebo ($P = 0.16$). 40 (8.6%) serious adverse events occurred in patients treated with N-acetylcysteine, and 31 (6.7%) serious adverse events occurred in patients treated with placebo ($P = 0.32$). Eight patients died during the study: three in the N-acetylcysteine group and five in the placebo group (Supplementary Table 8). No deaths were regarded by the investigators as being related to the study medication.

### Prespecified and non-prespecified exploratory analyses

The prespecified subgroup analyses revealed that N-acetylcysteine was associated with a significant reduction in the annual rate of total exacerbations and moderate or severe exacerbations among patients with mMRC score <2, CAT score <10, or GOLD stage 2 (Fig. 4). In patients with mMRC score ≥ 2, CAT score ≥ 10, and GOLD stage 1, there was a trend toward a lower annual rate of moderate or severe exacerbations, but the difference was not statistically significant (Fig. 4). In exploratory subgroup analyses that were not predefined, we found that N-acetylcysteine was associated with a greater benefit in the subgroup with ever smoker and COPD exacerbation in the year before the study started (Fig. 4).

### Discussion

In this trial, long-term treatment of high-dose N-acetylcysteine neither significantly reduced the annual rate of total exacerbations nor improved lung function in patients with COPD of GOLD stage 1–2. This study was registered on the Chinese Clinical Trial Registry (www.chictr.org.cn), number ChiCTR-IIR-17012604.

Reducing the social and economic burden of COPD requires a shift in intervention strategies, from reducing symptoms and

## CONSORT 2010 Flow Diagram

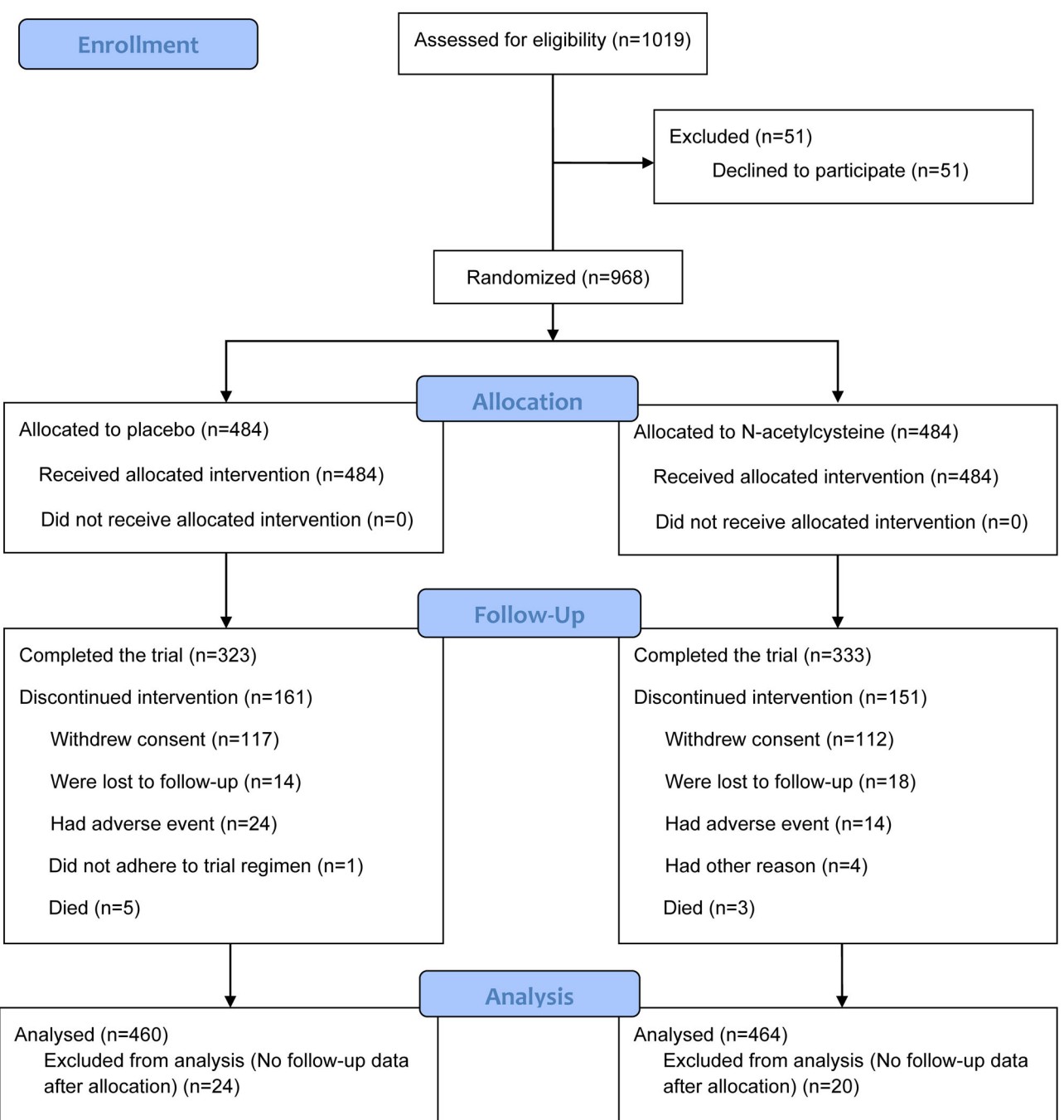

**Fig. 1 | Study flow diagram.** Patients who underwent randomization, were treated with at least one dose of *N*-acetylcysteine or placebo, and had available exacerbation assessment data at any planned follow-up visit were included in the full analysis set for exacerbation.

exacerbations in advanced disease to early intervention to slow disease progression in early-stage COPD. GOLD stage 1–2 COPD is not equivalent to early-stage COPD[18]. However, in this trial, we chose patients with COPD of GOLD stage 1–2 as the target population, mainly because lung function declines significantly faster in these patients than in those with advanced COPD[5]. Moreover, a considerable proportion of these patients experience exacerbation, and exacerbation has a more significant impact on disease progression than in patients with advanced COPD[6,8,9]. Only our previous study (Tie-COPD study) found that tiotropium could improve lung function, reduce acute exacerbation, and seem to reduce the annual decline in lung function in patients with COPD of GOLD stage 1–2[9]. It is thus necessary to identify more effective medications for the treatment of GOLD stage 1–2 COPD. *N*-acetylcysteine is a well-known medication that has both expectorant and antioxidant properties[10–13]. A previous study found that *N*-acetylcysteine could reduce the annual rate of exacerbations in

## Table 1 | Patients' baseline characteristics (full analysis set for exacerbation)

| Characteristic | N-acetylcysteine Group (N = 464) | Placebo Group (N = 460) |
|---|---|---|
| Age—year | 62.5 ± 8.4 | 62.6 ± 8.0 |
| Male sex —no. (%) | 414 (89.2) | 404 (87.8) |
| Body mass index—kg/m$^2$ | 22.7 ± 3.6 | 22.9 ± 3.4 |
| Smoking status—no. (%) | | |
| Never smoked | 71 (15.3) | 85 (18.5) |
| Former smoking | 135 (29.1) | 109 (23.7) |
| Current smoking | 258 (55.6) | 266 (57.8) |
| Smoking index — pack-year | 44.0 ± 25.6 | 43.6 ± 28.4 |
| Biomass use — no. (%) | 250 (53.9) | 258 (56.1) |
| Occupational exposure history — no. (%) | 226 (48.7) | 246 (53.5) |
| Family history of respiratory disease — no. (%) | 82 (17.7) | 73 (15.9) |
| Respiratory medication for COPD — no. (%) | 75 (16.2) | 62 (13.5) |
| LAMA alone | 37 (8.0) | 31 (6.7) |
| ICS and LABA | 27 (5.8) | 24 (5.2) |
| Traditional Chinese medicine | 7 (1.5) | 10 (2.2) |
| Xanthine | 9 (1.9) | 3 (0.7) |
| Asmeton | 3 (0.6) | 3 (0.7) |
| LABA alone | 4 (0.9) | 0 (0.0) |
| LABA and LAMA | 2 (0.4) | 1 (0.2) |
| ICS alone | 1 (0.2) | 0 (0.0) |
| COPD exacerbations in the previous year—no. (%) | 92 (19.8) | 108 (23.5) |
| Spirometric values at baseline | | |
| Before bronchodilator use | | |
| FEV$_1$—L | 1.98 ± 0.55 | 1.95 ± 0.55 |
| FEV$_1$ % of predicted value—% | 78.3 ± 16.7 | 78.1 ± 17.3 |
| FVC—L | 3.32 ± 0.79 | 3.26 ± 0.76 |
| FEV$_1$/FVC—% | 59.6 ± 7.5 | 59.3 ± 7.9 |
| After bronchodilator use | | |
| FEV$_1$—L | 2.08 ± 0.54 | 2.04 ± 0.54 |
| FEV$_1$ % of predicted value—% | 82.4 ± 15.8 | 82.0 ± 16.8 |
| FVC—L | 3.41 ± 0.75 | 3.38 ± 0.74 |
| FEV$_1$/FVC—% | 60.7 ± 7.5 | 60.2 ± 7.7 |
| Airflow reversibility—no. (%) | 69 (14.9) | 58 (12.6) |
| GOLD stage$^{\parallel}$—no. (%) | | |
| Mild (stage 1) | 261 (52.3) | 238 (47.7) |
| Moderate (stage 2) | 203 (47.8) | 222 (52.2) |
| mMRC dyspnea scale score | | |
| Mean score | 0.82 ± 0.78 | 0.80 ± 0.78 |
| Distribution—no. (%) | | |
| <2 | 392 (84.5) | 386 (83.9) |
| ≥2 | 72 (15.5) | 74 (16.1) |
| CAT score | | |
| Mean score | 8.1 ± 5.8 | 8.5 ± 6.1 |
| Distribution—no. (%) | | |
| <10 | 300 (64.7) | 279 (60.7) |
| ≥10 | 164 (35.3) | 181 (39.3) |

Data are n (%) or mean (standard deviation).
*COPD* chronic obstructive pulmonary disease, *ICS* inhaled corticosteroids, *LABA* long-acting β-agonist, *LAMA* long-acting muscarinic antagonist, *FEV1* forced expiratory volume in 1 s, *FVC* forced vital capacity, *GOLD* Global initiative for chronic obstructive lung disease, *mMRC* modified Medical Research Council. *CAT* chronic obstructive pulmonary disease assessment test.

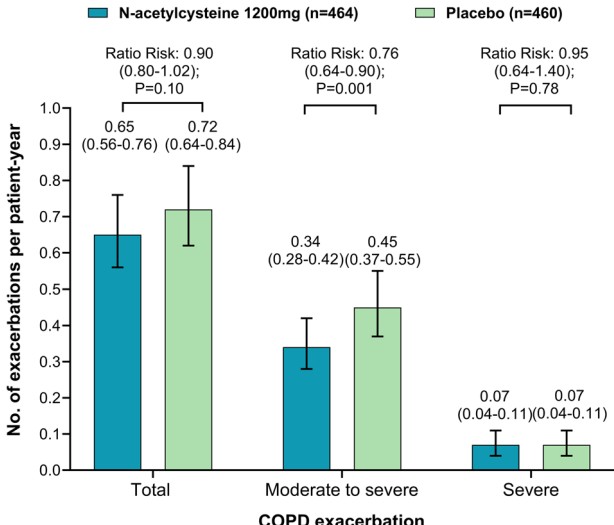

**Fig. 2 | Acute exacerbation of COPD during the study (full analysis set for exacerbation).** Data are mean ± standard error. COPD=chronic obstructive pulmonary disease. The relative risk was calculated using Poisson regression, with correction for exposure to the trial regimen, overdispersion, age, sex, body mass index, smoking status, COPD exacerbations in the previous year, COPD treatment at baseline and center. The error bars indicate 95% confidence intervals.

patients with advanced COPD[14,15,19]. Moderate-to-severe exacerbation also accelerated lung function decline in patients with GOLD stage 1–2 COPD[8]. Therefore, we hypothesize that long-term treatment with *N*-acetylcysteine may reduce the annual rate of exacerbations and ameliorate the annual decline in lung function in patients with GOLD stage 1–2 COPD. With this in mind, we set exacerbation and lung function as coprimary outcomes for patients with mild-to-moderate COPD in this clinical trial.

This study is one of the few randomized clinical trials in patients with GOLD stage 1–2 COPD and the first randomized clinical trial to investigate the efficacy and safety of long-term treatment with high-dose *N*-acetylcysteine in patients with GOLD stage 1–2 COPD. In this study, patients with GOLD stage 1–2 COPD were recruited from the community, with a mean FEV$_1$ of 78.2% of the predicted value. More than 50% of the patients had GOLD stage 1 COPD, more than 80% of the patients had an mMRC score of <2, and more than 60% of the patients had a CAT score of <10. Therefore, a considerable proportion of the patients in this study have mild disease status. However, we did not find evidence that *N*-acetylcysteine treatment can reduce the incidence of acute exacerbations or improve FEV$_1$ in patients with mild-to-moderate COPD.

In the analysis of secondary outcomes, we found that *N*-acetylcysteine reduced moderate-to-severe acute exacerbations in patients with mild-to-moderate COPD, similar to the findings of the PANTHON study, HIACE study, and BRONCUS study targeting patients with advanced COPD[14,15,19]. The 24% reduction in the moderate-to-severe exacerbations rate with *N*-acetylcysteine treatment has certain clinical significance, considering that previous COPD pharmacological trials design set a 15–25% reduction in the moderate-to-severe exacerbations rate. Nevertheless, a greater benefit was observed in patients with ever smoker, GOLD stage 2, and COPD exacerbation in the previous year at baseline. Interestingly, we also observed that *N*-acetylcysteine can reduce the annual rate of total exacerbations and the annual rate of moderate or severe exacerbations in subgroups of patients with mMRC score <2 and CAT score <10. We did not observe the benefit of *N*-acetylcysteine treatment in patients with mMRC ≥ 2 and CAT ≥ 10. The study might not have had enough participants (underpower) in certain subgroups with many respiratory symptoms to reliably assess *N*-acetylcysteine's effectiveness. Further study about the efficacy of

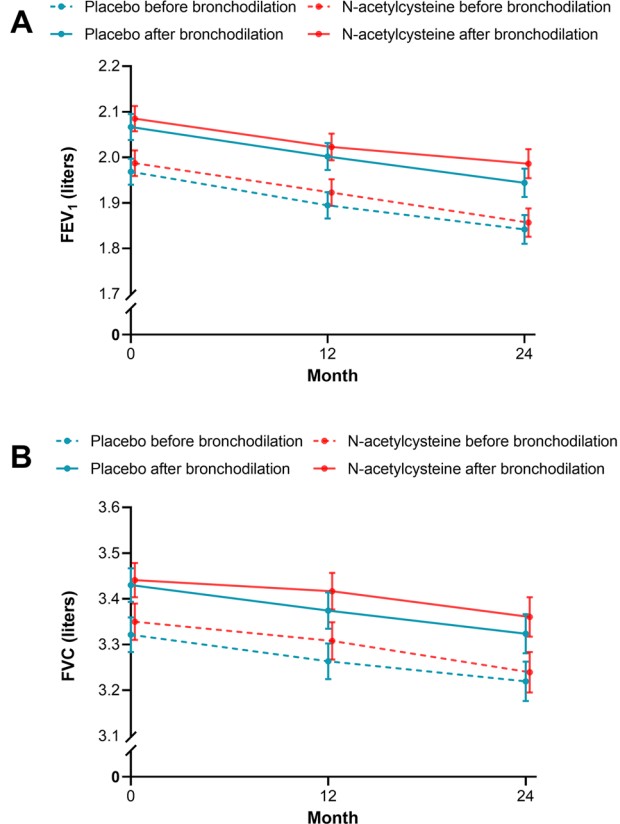

**Fig. 3 | Mean forced expiratory volume in 1 s (FEV₁) and forced vital capacity (FVC) before and after bronchodilator use over time (full analysis set for lung function).** Mixed-effects model for repeated measures was used to compare the differences in lung function before and after bronchodilator use at each visit between the two groups. The error bars indicate ± standard error. *N*-acetylcysteine did not significantly improve FEV₁ before bronchodilator use when compared with placebo at 24 months (mean difference: 16 ml, 95% confidence interval, −69 to 100 ml; *P* = 0.72) and at each visit (*P* > 0.05 for all comparisons) (**A**). *N*-acetylcysteine did not significantly improve FVC when compared with placebo at 12 or 24 months (*P* > 0.05 for all comparisons) (**B**).

*N*-acetylcysteine in patients with mMRC score ≥ 2 and CAT score ≥ 10 is still needed. Our findings help to guide pharmacotherapy in patients with GOLD stage 1–2 COPD and have important implications for clinical practice.

Our study revealed a statistically significant difference in the rate of moderate-to-severe exacerbation between the two groups, but not in the rate of total exacerbations. This may be explained as follows. First, the judgment and reporting of moderate-to-severe exacerbations are clear, while mild exacerbations may be underreported. For this reason, the most recent COPD clinical trials have used the annual rate of moderate-to-severe exacerbation as the primary outcome. Second, *N*-acetylcysteine can reduce moderate-to-severe exacerbation, but this is not as obvious as bronchodilators. *N*-acetylcysteine may reduce the severity of moderate-to-severe exacerbations, turning them into mild exacerbations, resulting in no significant between-group difference in the rate of total exacerbations.

This trial considered coprimary outcomes and two FASs were established. Since lung function was measured annually in this trial, some patients withdrew within the first year of the trial without available spirometry data at 12 or 24 months. Therefore, the number of patients in the FAS for lung function was smaller than that in the FAS for exacerbation. High-dose *N*-acetylcysteine did not significantly improve FEV₁ or FVC before or after bronchodilator use compared with placebo at 24 months. Moreover, it did not ameliorate the annual decline in FEV₁

and FVC before or after bronchodilator use in patients with GOLD stage 1–2 COPD. In contrast to these results, a small-sample clinical trial conducted by Pela and colleagues found that 600 mg *N*-acetylcysteine per day improved FEV₁ at 6 months in patients with moderate-to-severe COPD[20]. However, most clinical trials in patients with moderate-to-severe COPD or COPD with a history of exacerbation in the previous year have shown that *N*-acetylcysteine does not improve lung function[14,15,19,21], which is consistent with the results of this study. Further clinical trials in patients with COPD of GOLD stage 1–2 are still needed to screen for effective medications that can improve lung function and ameliorate the annual decline in lung function.

The incidence of adverse events and serious adverse events with *N*-acetylcysteine was low and similar to placebo, indicating that long-term use of high-dose *N*-acetylcysteine is well tolerated and safe.

The enrollment period for the patients in this study was from September 7, 2017, to August 29, 2019 and the last patient completed follow-up on January 3, 2022. Previous studies found that the strict lockdown was associated with the reductions in acute exacerbations of COPD, likely due to reduced transmission of respiratory virus infections in the Corona Virus Disease 2019 (COVID-19) pandemic[22–24]. A lower baseline exacerbation rate may have an impact on the exacerbation rate in this trial. However, COVID-19 is unlikely to have affected the results of this study for the following reasons. Firstly, the last patient was enrolled in this study before the presence of COVID-19. Therefore, the influence of COVID-19 on patient screening and enrollment in this study was minimal. Secondly, there were no reports of patients contracting COVID-19, including adverse events and serious adverse events, because the final follow-up date of this study was before the widespread presence of COVID-19 in China by the end of 2022. Thirdly, follow-up for the majority of patients was completed before the presence of COVID-19, and for a few patients, follow-up occurred after its emergence. After the presence of COVID-19, we coordinated with the participating centers to enhance communication with the patients and allowed limited telephone follow-ups and delivery of study medications by courier. Since this study is a double-blind clinical trial, the impact of COVID-19 on follow-up and the rates of exacerbations for both the *N*-acetylcysteine group and the placebo group should be consistent. It is unlikely that COVID-19 would significantly affect the study conclusions.

This study had some limitations. Firstly, the proportion of females included in this trial was lower than that of Chinese patients with mild-to-moderate COPD, which may affect the extrapolation of the results of this trial[3,4]. Secondly, we assumed *N*-acetylcysteine has an impact on lung function due to its anti-inflammatory and antioxidant effects[20]. However, this trial may have been underpowered to detect the effect of *N*-acetylcysteine on lung function due to the overestimation of the difference in lung function between the two groups during the calculation of the sample size. Third, the diagnosis of mild exacerbation is based on the new onset or worsening of respiratory symptoms and the addition of commonly used COPD medications at home, which may be underdiagnosed. We use diary cards to record daily symptoms and perform follow-up assessments every three months. This method is one of the best ways to identify acute exacerbations, especially the mild classification. It's worth noting that the likelihood of underdiagnosing mild acute exacerbations was same in the N-acetylcysteine group and the placebo group. The proportion of mild acute exacerbations in all acute exacerbations in this study was the same as in previous studies[9,25]. Therefore, the problem of underdiagnosis of mild exacerbations is unlikely to affect the conclusions of this study. Finally, the proportion of patients who withdrew from this study was slightly higher than in other clinical trials of advanced COPD but similar to that of our previous Tie-COPD study[9]. The dropout rate may be related to the high proportion of mild COPD, few symptoms, and a 2-year treatment period. Patients with mild COPD or few symptoms are less willing to treat and then are more likely to withdraw from the trial.

**Table 2 | Annual declines in lung function before and after bronchodilator use (full set analysis for lung function)**

| Variable | Decline per Year | | | P value | |
|---|---|---|---|---|---|
| | Placebo group | N-acetylcysteine group | Difference (95% CI)[a] | Unadjusted | Adjusted[b] |
| *FEV$_1$ (ml)* | | | | | |
| Before bronchodilator use | 63 ± 7 | 67 ± 7 | −4 (−23 to 16) | 0.69 | 0.68 |
| After bronchodilator use | 61 ± 7 | 56 ± 7 | 5 (−14 to 24) | 0.61 | 0.51 |
| *FVC (ml)* | | | | | |
| Before bronchodilator use | 51 ± 14 | 60 ± 13 | −9 (−46 to 28) | 0.63 | 0.54 |
| After bronchodilator use | 55 ± 13 | 48 ± 13 | 6 (−30 to 42) | 0.73 | 0.69 |
| *FEV$_1$/FVC* | | | | | |
| Before bronchodilator use | 0.98 ± 0.21 | 0.94 ± 0.20 | 0.03 (−0.54 to 0.61) | 0.91 | 0.80 |
| After bronchodilator use | 0.74 ± 0.21 | 0.75 ± 0.21 | −0.01 (−0.58 to 0.57) | 0.99 | 0.96 |
| *FEV$_1$ (% of predicted value)* | | | | | |
| Before bronchodilator use | 1.7 ± 0.3 | 1.8 ± 0.3 | −0.2 (−0.9 to 0.6) | 0.71 | 0.76 |
| After bronchodilator use | 1.5 ± 0.3 | 1.4 ± 0.3 | 0.2 (−0.6 to 0.9) | 0.70 | 0.54 |
| *FVC (% of predicted value)* | | | | | |
| Before bronchodilator use | 0.7 ± 0.5 | 1.2 ± 0.04 | −0.4 (−1.7 to 0.8) | 0.49 | 0.42 |
| After bronchodilator use | 0.9 ± 0.4 | 0.8 ± 0.4 | 0.05 (−1.1 to 1.2) | 0.93 | 0.79 |

A random-effects model was adopted, with the annual decline at individual time point as the random-effects variable, group assignment and individual baseline values as fixed-effect variables, and age, sex, body-mass index, baseline smoking status, and individual baseline values as the fixed-effect covariates.

Plus–minus values are means ± standard error per year, *FEV$_1$* forced expiratory volume in one second, *FVC* forced vital capacity.

[a]The difference was calculated as the value in the placebo group minus the value in the tiotropium group. Values may not sum as expected owing to rounding.

[b]P values were adjusted by the above-mentioned fixed-effect covariates, including age, sex, body-mass index, baseline smoking status, COPD exacerbations in the previous year, COPD treatment at baseline, center, and individual spirometric values at baseline (FEV$_1$, FVC, FEV$_1$/FVC, FEV$_1$ % of predicted value, and FVC % of predicted value before and after bronchodilator use).

A 35% dropout rate was considered when calculating the sample size. Moreover, no significant difference in the dropout rate and baseline characteristics was observed between the two groups. Therefore, the dropout rate is unlikely to have influenced the results of this study.

In summary, long-term treatment with high-dose N-acetylcysteine did not reduce the annual rate of total exacerbations and did not improve the lung function, but resulted in a lower annual rate of moderate or severe exacerbations than placebo in patients with COPD of GOLD stage 1–2. In addition, N-acetylcysteine was very well tolerated. It may be worth assessing the effect of N-acetylcysteine in mild-to-moderate COPD patients with chronic bronchitis phenotype and history of exacerbations within the previous 12 months, selecting moderate or severe exacerbations as the sole primary outcome measure. The effect of N-acetylcysteine in pre-COPD should be evaluated in future trials, especially in patients with normal spirometry measurements and chronic respiratory symptoms[26,27].

## Methods

We aimed to conduct a randomized controlled trial to investigate the efficacy and safety of long-term treatment with high-dose N-acetylcysteine (600 mg, twice daily) in patients with COPD of GOLD stage 1–2. Here, we report the methods of this randomized clinical trial.

### Study design

This is a multicentre, double-blind, parallel-group, placebo-controlled, randomized controlled trial at 24 centers in China from September 7, 2017, through January 3, 2022. Patients with mild-to-moderate COPD were randomized 1:1 to receive high-dose N-acetylcysteine (600 mg, twice daily) or placebo (twice daily) for two years. This trial was approved by the ethics committee of The First Affiliated Hospital of Guangzhou Medical University, Huizhou First Hospital, Tianjin Medical University General Hospital, The Second People's Hospital of Hunan Province, The Third Affiliated Hospital of Guangzhou Medical University, Liwan Hospital, The Second Xiangya Hospital of Central South University, Fudan University Pudong Medical Center, The First Hospital of China Medical University, Tongji Hospital of Tongji Medical College, The First Hospital of Wenzhou Medical University, Shenzhen People's Hospital, The First Hospital of Guangxi Medical University,

Peking University Third Hospital, Hunan Provincial People's Hospital, Zhongshan Hospital of Fudan University, Wengyuan County People's Hospital, Lianping County People's Hospital, Ruijin Hospital of Shanghai Jiao Tong University School of Medicine, Guangzhou First People's Hospital, The Second Affiliated Hospital of Zhejiang University School of Medicine, Sir Runrun Shaw Hospital, The First Affiliated Hospital of Jinan University, and The Affiliated Hospital of Guangdong Medical University according to the requirements of Chinese clinical trial guidelines. All patients signed informed consent before enrollment. The study design of this study was published and the study protocol and statistical analysis plan was provided in Supplement material[28]. This study was registered on the Chinese Clinical Trial Registry (www.chictr.org.cn), number ChiCTR-IIR-17012604.

### Patients

Eligible patients were 40-80 years old and GOLD stage 1–2 COPD (a ratio of FEV$_1$ to FVC ratio of <0.70 and an FEV$_1$ of ≥ 50% predicted value after bronchodilator use), had chronic respiratory symptoms and/or risk factors for COPD (e.g., smoking, biofuel exposure, air pollution)[9,10]. Additional inclusion criteria included stable COPD (no acute exacerbations in the 4 weeks prior to the screening period), capability to communicate verbally or through written documents and provide informed consent, and willingness and capability to undergo relevant ancillary examinations related to this study. The exclusion criteria were (1) COPD exacerbation within four weeks prior to the screening period; (2) long-term treatment with N-acetylcysteine for more than three months prior to the screening period; (3) patients with a clinical diagnosis of lung cancer, bronchiectasis, pneumoconiosis, asthma, interstitial lung disease, or other single restricted ventilation; (4) significant diseases other than COPD (A significant disease was defined as a disease or condition which, in the opinion of the investigator, may put the patient at risk because of participation in the study or may influence either the results of the study or the patients' ability to participate in the study); (5) patients with clinically significant abnormal baseline blood routine examination, blood biochemistry or urinary analysis if the abnormality defines a significant disease as defined in exclusion criteria No. 4; (6) severe cardiovascular, neural, hepatic, renal, and hematologic diseases or malignancies that may interfere

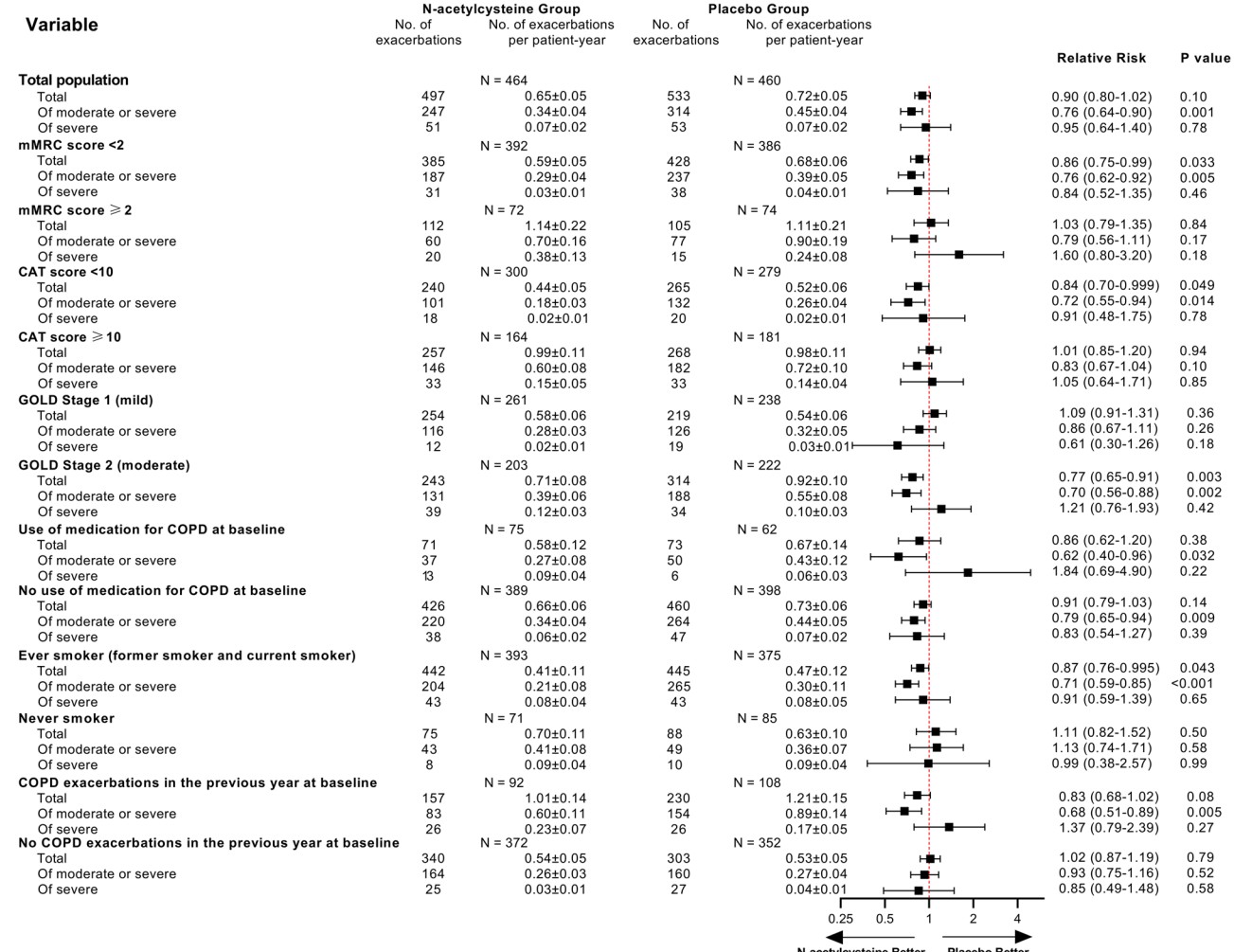

**Fig. 4 | Forest plot of acute exacerbation of COPD according to subgroups (full analysis set for exacerbation).** COPD=chronic obstructive pulmonary disease. GOLD=Global Initiative for Chronic Obstructive Lung Disease. CI=confidence interval. mMRC=modified Medical Research Council. CAT=chronic obstructive pulmonary disease assessment test. Relative risk and 95% confidence intervals are shown and the error bars indicate 95% confidence intervals. The relative risk was calculated using Poisson regression, with correction for exposure to the trial regimen, overdispersion, age, sex, body mass index, smoking status, COPD exacerbations in the previous year, COPD treatment at baseline and center.

with the operation of the study; (7) known moderate to severe renal impairment, judged by the investigator or creatinine clearance ≤50 ml/min; (8) patients with phenylketonuria; (9) known hypersensitivity or intolerance to trial drugs; (10) patients with severe gastric ulcer or intestinal malabsorption; (11) patients with active pulmonary tuberculosis; (12) patients with life-threatening pulmonary embolism, α1-antitrypsin deficiency, or cystic fibrosis; (13) history of pneumonectomy; (14) pregnancy, lactation, or potential of pregnancy; (15) long-term oxygen therapy, frequent use of corticosteroids orally or intravenously (prednisone >10 mg/d), or long-term use of antibiotics; (16) planned hospitalization or blood donation during the trial; (17) history of chronic alcohol or drug abuse, or any other conditions that may impact compliance; (18) involvement in other clinical studies at the same time; (19) patients who need long-term oxygen therapy and rehabilitation in the next 2 years.[28]

### Randomization and masking
The randomization was centralized and stratified by center, with block sizes of four. Investigators and centers maintained emergency sealed opaque envelopes for each numbered drug, containing the randomization code for a given patient, for emergency unblinding only in the event of a serious life-threatening adverse event. The principal

investigator, physicians, patients, and statisticians were blinded to the drug allocation when the study was conducted. Zhejiang Jinhua Pharmaceutical (Hangzhou, China) manufactured and provided N-acetylcysteine and matched placebo. The matched placebo was identical to N-acetylcysteine in shape, color, size, and packaging but did not contain any active ingredients.

### Procedures
The screening period was one week, and clinical visits were then scheduled every three months. Patients were asked to record their daily medication usage, respiratory symptoms, changes in respiratory symptoms, and rescue medication (salbutamol) usage. If patients experienced new or worsening respiratory symptoms, they were instructed to document the medication used for managing exacerbations and contact the investigator as soon as possible. The incidence of acute exacerbations of COPD was confirmed by the investigator based on a comprehensive assessment of patient diary card records or documented hospital visits. COPD exacerbation was defined as the appearance or worsening of at least two major symptoms (cough, expectoration, purulent sputum, wheezing, or dyspnoea) persisting for at least 48 h following exclusion of cardiac insufficiency, pulmonary embolism, pneumothorax, pleural effusion, or cardiac arrhythmia

has been ruled out[9,29,30]. The severity of exacerbation was graded by the investigator according to the following categories: severe (requiring hospitalization); moderate (requiring outpatient or emergency department visits for increasing medication, including antibiotics and/or systemic glucocorticoids); and mild (adding commonly used COPD medications at home)[9,29,30]. If patients self-administer antibiotics and/or systemic corticosteroids at home during an exacerbation, it is also considered a moderate exacerbation. The mMRC dyspnoea scale and the CAT score were assessed at each visit[31,32]. Blood samples were taken from a subgroup at screening and 24 months, which were analyzed in a blinded manner at a central laboratory after study completion.

Spirometry was measured within 2 h for each visit (at screening, baseline, 12 months, and 24 months) by the same trained technicians according to the international standards[33,34]. Short-acting bronchodilators were not permitted within 6 h before lung function tests, while long-acting bronchodilators were not permitted within 24 h. Airway reversibility was assessed via bronchodilation with inhalation of 400 µg salbutamol (Ventolin, Glaxo Wellcome, UK) at 20 min after baseline spirometry. The predicted $FEV_1$ was derived from European Coal and Steel Community 1993 reference values and by the application of conversion factors for Chinese individuals (0.95 in men and 0.93 in women)[35,36].

## Outcomes

The coprimary outcomes were the annual rate of total (mild, moderate, and severe) exacerbations and the difference in $FEV_1$ before bronchodilator use at 24 months from baseline. The secondary outcomes included: the annual rate of moderate or severe exacerbations; the annual rate of severe exacerbations; the time to the first total exacerbation; the time to the first moderate-to-severe exacerbation; the time to the first severe exacerbation; the difference in $FEV_1$ after bronchodilator use at 24 months from baseline; FVC before and after bronchodilator use at 24 months from baseline; annual decline in $FEV_1$, $FEV_1$ % of predicted value, FVC, FVC % of predicted value, and $FEV_1$/FVC before and after bronchodilator use; mMRC score; CAT score; and adverse events. We performed prespecified subgroup analyses according to the mMRC score (<2 or ≥2), CAT score (<10 or ≥10), and GOLD stage (1 [mild] or 2 [moderate]). We also performed additional non-pre-specified exploratory subgroup analyses according to smoking status (never smoker or ever smoker), use of medication for COPD at baseline (Yes or No), and COPD exacerbations in the previous year at baseline (Yes or No). All secondary, prespecified, and non-pre-specified exploratory subgroup analyses should be interpreted as exploratory. These analyses have not been adjusted for multiple comparisons and are excluded from formal hypothesis testing.

## Statistical analysis

The sample size was calculated based on the coprimary outcomes. First, referring to the Tie-COPD study and PANTHEON study[9,14], we calculated the sample size using Poisson regression analysis to detect a difference of 0.23 per patient-year between the *N*-acetylcysteine group (0.27 per patient-year) and the placebo group (0.50 per patient-year) in the annual rate of exacerbations, with a two-sided significance level of 5% and a power of 90% based on an anticipated withdrawal rate of 35%. Thus, a minimum of 230 patients needed to be randomized per group. There are currently no studies on how much of a reduction in exacerbation rates would constitute a minimal clinically important difference[37]. Our hypothesized difference in exacerbation rate of 0.23 is equivalent to 46% fewer exacerbations in the treatment group compared with placebo. Therefore, we believe that a reduction of 0.23 per patient-year exacerbations is clinically significant, considering that previous clinical trials of COPD drugs were designed with a 15–25% reduction in exacerbation rate. Second, referring to the study by Pela

and colleagues[20], we calculated the sample size to detect the difference in $FEV_1$ before bronchodilator use between the *N*-acetylcysteine group and the placebo group, assuming a difference of 100 ml and a standard deviation of 450 ml at month 24, with a two-sided significance level of 5% and a power of 80% based on an anticipated withdrawal rate of 35%. A minimum of 489 patients needed to be randomized per group. The sample size calculation for the selection of two coprimary outcomes was not adjusted for multiple comparisons. We chose the largest sample size calculated for the two coprimary outcomes for this study. Combining the calculation results of the two sample sizes, at least 489 patients needed to be randomized in each group. The statistical analysis plan, which was finalized before the data were finalized, specified the content of the statistical analysis in detail.

A predefined analysis of exacerbation rate was performed using the Poisson regression model, with adjustment for exposure to treatment doses, age, sex, body mass index, smoking status, COPD exacerbations in the previous year, COPD treatment at baseline, and center. It was found that the number of acute exacerbations in this study had an over-dispersed distribution during data analysis, so we further adjusted the deviance over-dispersed in the Poisson regression model[25,38,39]. The rate of COPD exacerbations per patient per year was estimated using the Poisson regression model. Time to the first exacerbation was evaluated using the Cox proportional hazards regression model. A mixed-effects model for repeated measures was used to compare the differences in $FEV_1$ and FVC before and after bronchodilator use at each visit between the two groups. The value measured at each visit was used as the dependent variable. The fixed effects in the model included the treatment method, patients' baseline lung function values, and follow-up visits (treated as categorical variables), and the random effects included the interaction of treatment with follow-up and center. We used a random coefficient regression model to compare the differences in the annual decline in $FEV_1$, $FEV_1$ % of predicted value, FVC, FVC % of predicted value, and $FEV_1$/FVC between the two groups. The model regression coefficient was used to represent the annual decline in the treatment effect over time, and the maximum-likelihood algorithm for imputing missing data. A mixed-effects model for repeated measures was used to compare the CAT score at each time point after treatment between the two groups. Cochran's and Mantel-Haenszel test was used to compare the change in the mMRC score. The number of adverse events for both groups was displayed. Except for using the maximum likelihood algorithm to estimate missing data in the random coefficient regression model, no imputation was performed for missing data in the remaining analyses.

Patients who underwent randomization were treated with at least one dose of study medication, and had available post-baseline data on exacerbation were included in the FAS for exacerbation, and those who had available spirometry data at 12 months or 24 months were included in the FAS for lung function. All randomized patients were included in the safety analysis set. Statistical analyses were performed using SAS 9.4 (SAS Institute, USA), and all reported P values are two-sided.

## Reporting summary

Further information on research design is available in the Nature Portfolio Reporting Summary linked to this article.

## Data availability

Deidentified clinical data for individual patients supporting the results of this manuscript and supplementary information will be available, for up to 15 years after this paper's publication. Applications for access to data should be made in writing to Prof. Pixin Ran (pxran@gzhmu.edu.cn) and data will be provided within 1 month after approved by investigators and data transfer agreement has been signed. Source data are provided with this paper.

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

## Acknowledgements

The authors thank all trial staff, the participants, and their families who participated in this clinical trial. We would like to express our gratitude to Dr. Zihui Wang for providing valuable feedback on the manuscript revision. This study was supported by grants from the Plan on Enhancing Scientific Research in Guangzhou Medical University (GMUCR2024-

01012), the National Key Research and Development Program of the 13th National 5-Year Development Plan (2016YFC1304101), the Local Innovative and Research Teams Project of Guangdong Pearl River Talents Program (2017BT01S155), the foundation of Guangzhou National Laboratory (SRPG22-016 and SRPG22-018), and the Clinical and Epidemiological Research Project of State Key Laboratory of Respiratory Disease (SKLRD-L-202402). The funding providers and Zhejiang Jinhua Pharmaceutical (Hangzhou, China) had no role in the study design, implementation, monitoring, statistical analysis, interpretation, writing, and publication of the manuscript.

## Author contributions

Yumin.Z., F.W., Z.S., J.C., J.T., W.Y., L.W., F.L., S.C., Y.S., Z.W., H.Z., Y.C., Y.F., Z.H., C.C., Y.J, S.C., C.Y., S.Y., and H.T. contributed equally to this work. P.R. and Yumin. Z. had the idea for and designed the study. P.R. and N.Z. supervised the study. F.W. and Yumin. Z. did the statistical analysis. P.R., N.Z., Yumin.Z., F.W., Z.S., J.C., J.T., W.Y., L.W., F.L., Shan.C., Y.S, Z.W., H.Z., Y.C., Y.F., Z.H., C.C., Y.J, Shujing.C., C.Y., S.Y., H.T., Q.C., Z.Z., Y.Y., Yong.Z., S.L., Z.D., P.H., Yunzhen.Z., X.L., H.Z., J.G., W.L., G.H., C.L., L.S., Z.L., J.H, and D.Z. contributed to the acquisition, analysis, or interpretation of data. F.W., Yumin. Z., and P.R. wrote the draft manuscript. P.R., N.Z., Yumin.Z., F.W., Z.S., J.C., J.T., W.Y., L.W., F.L., Shan.C., Y.S, Z.W., H.Z., Y.C., Y.F., Z.H., C.C., Y.J, Shujing.C., C.Y., S.Y.,.,H.T., Q.C., Z.Z., Y.Y., Yong.Z., S.L., Z.D., P.H., Yunzhen.Z., X.L., H.Z., J.G., W.L., G.H., C.L., L.S., Z.L., J.H, and D.Z. revised the manuscript and approved the final version before submission. F.W., P.R., and Yumin.Z. affirm that the manuscript is an honest, accurate, and transparent account of the study being reported; that no important aspects of the study have been omitted; and that any discrepancies from the study as planned (and, if relevant, registered) have been explained.

## Competing interests

All authors declare: no support from any organization for the submitted work; no financial relationships with any organizations that might have an interest in the submitted work in the previous three years; no other relationships or activities that could appear to have influenced the submitted work.

## Ethical approval

This trial was approved by the ethics committee at each hospital according to the requirements of Chinese clinical trial guidelines. All patients provided written informed consent before enrollment.

## Additional information

A complete list of the investigators in the *N*-acetylcysteine in Mild-to-moderate Chronic Obstructive Pulmonary Disease in China trial is provided in the Supplement Material.

Yumin Zhou[1,2,27], Fan Wu [1,2,27], Zhe Shi[3,27], Jie Cao[4,27], Jia Tian[5,27], Weimin Yao[6,27], Liping Wei[7,27], Fenglei Li[8,27], Shan Cai[9,27], Yao Shen[10,27], Zanfeng Wang[11,27], Huilan Zhang[12,27], Yanfan Chen[13,27], Yingyun Fu[14,27], Zhiyi He[15,27], Chun Chang[16,27], Yongliang Jiang[17,27], Shujing Chen[18,27], Changli Yang[19,27], Shuqing Yu[20,27], Heshen Tian[1,27], Qijian Cheng[21], Ziwen Zhao[22], Yinghua Ying[23], Yong Zhou[24], Shengming Liu[25], Zhishan Deng[1], Peiyu Huang [1], Yunzhen Zhang[3], Xiangwen Luo[20], Haiyan Zhao[4], Jianping Gui[5], Weiguang Lai[26], Guoping Hu[7], Cong Liu[9], Ling Su[10], Zhiguang Liu[17], Jianhui Huang[20], Dongxing Zhao[1], Nanshan Zhong[1,2] & Pixin Ran [1,2] ✉, On behalf of China N-acetylcysteine in Mild-to-moderate COPD Study Group

[1]State Key Laboratory of Respiratory Disease & National Clinical Research Center for Respiratory Disease & National Center for Respiratory Medicine & Guangzhou Institute of Respiratory Health, The First Affiliated Hospital of Guangzhou Medical University, Guangzhou Medical University, Guangzhou, China. [2]Guangzhou National Laboratory, Bio-land, Guangzhou, China. [3]Huizhou First Hospital, Huizhou, China. [4]Tianjin Medical University General Hospital, Tianjin, China. [5]The Second People's Hospital of Hunan Province, Changsha, China. [6]Department of Respiratory and Critical Care Medicine, The Second Affiliated Hospital of Guangdong Medical University, Zhanjiang, China. [7]The Third Affiliated Hospital of Guangzhou Medical University, Guangzhou, China. [8]Liwan Central Hospital of Guangzhou, Guangzhou, China. [9]The Second Xiangya Hospital of Central South University, Changsha, China. [10]Department of Pulmonary and Critical Care Medicine, Shanghai Pudong Hospital, Fudan University Pudong Medical Center, Shanghai, China. [11]The First Hospital of China Medical University, Shenyang, China. [12]Department of Respiratory and Critical Care Medicine, Tongji Hospital, Tongji Medical College, Huazhong University of Science and Technology, Wuhan, China. [13]The First Hospital of Wenzhou Medical University, Wenzhou, China. [14]Shenzhen People's Hospital,

Shenzhen, China. [15]The First Hospital of Guangxi Medical University, Nanning, China. [16]Peking University Third Hospital, Beijing, China. [17]Hunan Provincial People's Hospital, Changsha, China. [18]Department of Pulmonary and Critical Care Medicine, Zhongshan Hospital, Fudan University, Shanghai, China. [19]Wengyuan County People's Hospital, Shaoguan, China. [20]Lianping County People,s Hospital, Lianping County Hospital of Traditional Chinese Medicine, Heyuan, China. [21]Ruijin Hospital, Shanghai Jiao Tong University School of Medicine, Shanghai, China. [22]Guangzhou First People's Hospital, Guangzhou, China. [23]Key Laboratory of Respiratory Disease of Zhejiang Province, Department of Respiratory and Critical Care Medicine, The Second Affiliated Hospital of Zhejiang University School of Medicine, Hangzhou, China. [24]Sir Run Run Shaw Hospital, Zhejiang University School of Medicine, Hangzhou, China. [25]Department of Respiratory and Critical Care Medicine, The First Affiliated Hospital of Jinan University, Guangzhou, China. [26]Department of Respiratory and Critical Care Medicine, The Affiliated Hospital of Guangdong Medical University, Zhanjiang, China. [27]These authors contributed equally: Yumin Zhou, Fan Wu, Zhe Shi, Jie Cao, Jia Tian, Weimin Yao, Liping Wei, Fenglei Li, Shan Cai, Yao Shen, Zanfeng Wang, Huilan Zhang, Yanfan Chen, Yingyun Fu, Zhiyi He, Chun Chang, Yongliang Jiang, Shujing Chen, Changli Yang, Shuqing Yu, Heshen Tian. A list of authors and their affiliations appears at the end of the paper.
✉e-mail: pxran@gzhmu.edu.cn

## China N-acetylcysteine in Mild-to-moderate COPD Study Group

Yumin Zhou[1,2,27], Fan Wu [1,2,27], Zhe Shi[3,27], Jie Cao[4,27], Jia Tian[5,27], Weimin Yao[6,27], Liping Wei[7,27], Fenglei Li[8,27], Shan Cai[9,27], Yao Shen[10,27], Zanfeng Wang[11,27], Huilan Zhang[12,27], Yanfan Chen[13,27], Yingyun Fu[14,27], Zhiyi He[15,27], Chun Chang[16,27], Yongliang Jiang[17,27], Shujing Chen[18,27], Changli Yang[19,27], Shuqing Yu[20,27], Heshen Tian[1,27], Qijian Cheng[21], Ziwen Zhao[22], Yinghua Ying[23], Yong Zhou[24], Shengming Liu[25], Zhishan Deng[1], Peiyu Huang [1], Yunzhen Zhang[3], Xiangwen Luo[20], Haiyan Zhao[4], Jianping Gui[5], Weiguang Lai[26], Guoping Hu[7], Cong Liu[9], Ling Su[10], Zhiguang Liu[17], Jianhui Huang[20], Dongxing Zhao[1], Nanshan Zhong[1,2], Pixin Ran [1,2]✉, Zhuxiang Zhao[22], Sha Liu[1], Zihui Wang[1], Longhui Tang[1], Jinzhen Zheng[1], Hailin Zhou[26], Yuliang Cai[26], Yu Zhang[26], Haiqing Li[7], Ping Chen[9], Fen Jiang[9], Suizheng Cai[10], Jian Zhou[10], Hong Xia[10], Meijia Wang[12], Jiaojiao Chu[12], Xueding Cai[13], Yazhen Li[14], Yanfei Bin[15], Lingshan Zeng[20], Xiangwen Luo[20], Haiqin Zhang[21], Hua He[22], Yichen He[23], Wenxia Zhou[24] & Li Chen[25]

