## [Peer Review File · Nature Communications]

Author's Response to Reviewers

Reviewer 1

Comment 1: *Remarks to the Author: The authors should be applauded for tackling a challenging subject – therapeutic trials in mild to moderate COPD. They conclude there is no effect of NAC on the co-primary endpoints.*

Reply: Thank you for your kind comments. Patients with mild-to-moderate COPD experienced a more rapid decline in lung function than patients with advanced COPD and a certain number of acute exacerbations. Exacerbation in mild-to-moderate COPD has a more serious impact on lung function decline than moderate-to-severe exacerbation in advanced COPD. It is urgent to alleviate exacerbation and lung function decline in patients with mild-to-moderate COPD. However, up to now, evidence for the treatment of patients with mild-to-moderate COPD is limited. Patients with mild-to-moderate COPD have fewer respiratory symptoms and are less likely to seek medical attention, especially those with mild COPD. Therefore, conducting clinical trials specifically targeting patients with mild-to-moderate COPD poses significant challenges. We appreciate your acknowledgement of our work.

Changes in the text: Without modification.

Comment 2: *Major queries: The study should be described as negative. The*

exacerbation data for the primary endpoint are such. The interpretation of potential of subsets of exacerbations should be toned down.

Reply: Thank you for your kind comments and instructions. We agree with your comment that the primary endpoint of the study should be described as a negative outcome in the manuscript. We have made significant revisions to the abstract and discussion sections accordingly. Currently, large clinical trials on COPD primarily focus on moderate-to-severe exacerbations as the primary endpoint, rather than total exacerbations. Therefore, although moderate-to-severe exacerbations were considered as secondary endpoints in our study, we still emphasized their importance. Based on your and the editor's suggestions, we have made extensive revisions to ensure that our study draws conclusions based on the primary research endpoint.

Changes in the text: We have modified our manuscript as advised (see Abstract and Discussion section in the revised manuscript).

Comment 3: *It is surprising that the authors do not mention the potential impact of the COVID-19 pandemic on the rates or nature of exacerbations. This is particularly relevant given the strict lockdowns in their country during part of the conduct of the study.*

Reply: The enrollment and follow-up period for the patients in this study was from September 7, 2017, to January 3, 2022. The impact of COVID-19 on this study was inevitable. However, we believe that COVID-19 is unlikely to have affected the results of this study for the following reasons. Firstly, the last patient was enrolled in this study

before the outbreak of COVID-19. After the widespread occurrence of COVID-19 in China, we obtained approval from the ethics committee to terminate patient enrollment. Therefore, the influence of COVID-19 on patient screening and enrollment in this study was minimal. Secondly, because the final follow-up date of this study was after the large-scale containment of COVID-19 in China by the end of 2022, there were no reports of patients being infected with COVID-19, including adverse events and serious adverse events. Thirdly, follow-up for the majority of patients was completed before the arrival of COVID-19, and for a few patients, follow-up occurred after its emergence. After the onset of COVID-19, we coordinated with the participating centers to enhance communication with the patients and allowed limited telephone follow-ups and delivery of study medications by courier. Since this study is a double-blind clinical trial, the impact of COVID-19 on follow-up and the rates of exacerbations for both the N-acetylcysteine group and the placebo group should be consistent. It is unlikely that this would significantly affect the study conclusions.

Changes in the text: We have modified our manuscript as advised (see Pages 12-13, lines 270-288 in the revised manuscript).

Comment 4: *The nature of the daily diary used to define exacerbation events is not described and this may have strong influence on ascertainment of exacerbation like episodes.*

Reply: We have supplemented the description of patient diary card reporting in the methods section according to the study protocol. Patients were asked to record their

daily medication usage, respiratory symptoms, changes in respiratory symptoms, and rescue medication usage. If patients experienced new or worsening respiratory symptoms, patients were instructed to document the medication used for managing exacerbations and contact the investigator as soon as possible. The incidence of acute exacerbations of COPD was confirmed by the investigator based on a comprehensive assessment of patient diary card records or documented hospital visits. Participants included in this study were followed up once every three months. This method, combined with patient diary cards and written medical records, is currently the most commonly used approach to determine the occurrence of acute exacerbations in patients, providing a comprehensive assessment of acute exacerbation evaluation (Lancet Respir Med. 2014;2(3):187-194. N Engl J Med. 2017;377(10):923-935. N Engl J Med. 2023;389(3):205-214.).

Changes in the text: We have modified our manuscript as advised (see Page 17, lines 374-380 in the revised manuscript).

Comment 5: *There is significant drop out which limits interpretation, particularly of the lung function data. I realize that the authors anticipated a 35% drop out rate over the two year duration of the study. Nevertheless examination of Extended Data Table 3 suggests differences in disease severity between dropouts and the main population as well as differences between the NAC and placebo control groups.*

Reply: The proportion of patients who withdrew from this study was slightly higher than in other clinical trials of advanced COPD, but similar to that of our previous Tie-

COPD study. The dropout rate may be related to the high proportion of mild COPD, few symptoms, and 2-year treatment period. Patients with mild COPD or few symptoms are less willing to treat and then are more likely to withdraw from the trial. A 35% dropout rate was considered when calculating the sample size. Moreover, no significant difference in the dropout rate was observed between the two groups. Therefore, the dropout rate is unlikely to have influenced the results of this study. We have added this part of the description to the limitations of the discussion. It is common to see a slight improvement in lung function in the N-acetylcysteine group compared to the placebo group among the patients who discontinued the study, just as in our previous Tie-COPD study (N Engl J Med. 2017;377(10):923-935.). If patients with more severe COPD receive placebo treatment, they are more likely to discontinue the study compared to those who receive N-acetylcysteine or placebo treatment. This indirectly suggests that patients subjectively perceive the N-acetylcysteine treatment as effective. The lung function of the two groups of COPD patients included in this study was similar, and adjustments were made for confounding factors when analyzing acute exacerbations and differences in lung function. Therefore, this is unlikely to affect the study conclusions.

Changes in the text: Without modification.

Comment 6: *The nature of the patient population is quite unusual given restraints on allowed concomitant medications allowed (e.g. inhaled steroids excluded). Despite respiratory symptoms and prior exacerbations in a proportion of individuals there is*

remarkably little concomitant medication used. This limits the generalizability of the study results. I realize that in point #1 I acknowledge the negative primary results but this comment supports the need to temper any sub analysis of exacerbation endpoints.

Reply: When designing this study, we took into account the concomitant medication use of patients with mild-to-moderate COPD. At that time, the GOLD recommendations allowed the use of bronchodilators for COPD patients with minimal symptoms and a low risk of exacerbation. Therefore, we did not restrict the continued use of bronchodilators by patients with mild-to-moderate COPD upon entry into this study, but we did restrict the use of inhaled corticosteroids. The current GOLD 2024 recommendations still support this approach. Additionally, considering the limited treatment utilization among patients with mild-to-moderate COPD in real-world and community settings, the conduct of this study aligns with the current reality of COPD community management. Therefore, we believe that limiting the use of inhaled corticosteroids in this study is unlikely to affect the generalizability of our study findings.

Changes in the text: Without modification.

Comment 7: *Minor queries:*

1) *Line 73 – should be ‘... COPD is....’*

2) *Line 101 – delete ‘the’ in ‘...alleviate the moderate...’*

3) *Lines 110-111 – Reword to ‘The efficacy of N-acetylcysteine....COPD is still unclear.*

A dose-response..., we hypothesized....’

- 4) Line 116 – reword to ‘...and lung function...’
- 5) Line 127 – reword to ‘...completed follow-up’
- 6) Line 129 – reword to ‘...p=0.49). There was no ...’
- 7) Line 132 – note major comment #4 above. I don’t completely accept this interpretation.
- 8) Line 133 and Extended Data Table 2 – there seems to be an error in this table in the post-bronchodilator FEV1 of 2.68. I suspect this should be 2.08.
- 9) Line 137 – see major comment #4 above.
- 10) Line 142 – how do you explain measurable circulating NAC in placebo patients?
Was there off-study use that could have influenced study results?
- 11) Line 153 – see major point #1 above.
- 12) Lines 188-189 – This needs to be tempered given the negative nature of the predefined co-primary endpoint.
- 13) Lines 216-220 – on the other hand a significant proportion had respiratory symptom burden and prior exacerbation history – see major point #5 above.
- 14) Lines 268-271 – see major point # 4 above.
- 15) Line 364 – reword to ‘...centres maintained...’
- 16) Line 377 – this highlights major point # 3 above. Wheezing has generally been poorly associated with exacerbation ascertainment in prior studies.

Reply: Thank you for your kind comments and instructions. We have made revisions to address all the points mentioned in your suggestions.

We assessed the circulating N-acetylcysteine levels to confirm its administration

in the treatment group and to verify if these levels were significantly higher compared to the placebo group, ensuring adequate blood drug concentration. The difference in circulating N-acetylcysteine concentration is not an endpoint for assessing the efficacy and safety of this study but rather serves as a general clinical characteristic included in the analysis. Additionally, the measurement of circulating N-acetylcysteine concentration was not intended to exclude the possibility of off-study use of N-acetylcysteine. During study enrollment and at each follow-up visit, we emphasized to patients that if they experience new or increased respiratory symptoms, they should not use N-acetylcysteine for treatment outside of the study.

The initial definition of acute exacerbations of COPD included symptoms of wheezing (Ann Intern Med. 1987;106(2):196-204.). However, there is still some controversy regarding the definition of acute exacerbations of COPD (Am J Respir Crit Care Med. 2021;204(11):1251-1258.). In order to maintain comparability with previous studies, this trial adopted the same definition of COPD exacerbations used in the UPLIFT study (N Engl J Med. 2008;359(15):1543-1554.) and Tie-COPD study (N Engl J Med. 2017;377(10):923-935.).

Changes in the text: We have modified our manuscript as advised.

Reviewer 2

Comment 1: *Remarks to the Author: Summary of key results, originality and significance: The authors highlight the immense burden of COPD on patients and health services globally, and that the whilst the rate of lung function decline is most rapid in mild COPD, most therapeutic trials target patients with moderate to severe impairment. This study addresses an area of urgent clinical need, though to limit progression of mild (and early) COPD, reducing exposure to causative agents requires greater attention in the background and discussion. N-acetylcysteine is well tolerated, relatively inexpensive, and has not previously been studied in this target population. Treatment was associated with a modest reduction in moderate to severe exacerbations, but not total exacerbations. Furthermore, the reduction in moderate to severe exacerbations did not translate into better health-related quality of life or lung function. Ever-smoker status, moderate COPD and a history of exacerbations was associated with greater response. Perhaps surprisingly, a greater benefit was reported in patients with lower symptom scores, although the authors highlight that only a small proportion of patients had higher symptom scores, and the study may simply have been underpowered. Unfortunately, there is no health economic data available, limiting the scope to implement the findings of this study within healthcare systems.*

Reply: Thank you for your kind comments. We respond point-to-point to your comments below.

Changes in the text: Without modification.

Comment 2: *In some countries patients are given rescue antibiotics and steroids to keep at home, with education on exacerbation recognition and self-management. The definition of mild exacerbations (“adding commonly used COPD medications at home”) could be clearer, as use of rescue steroids and antibiotics would elevate the exacerbation to moderate severity.*

Reply: Thank you for your kind comments and suggestions. Due to increasingly strict regulations on antibiotics and systemic corticosteroids in China, it is difficult for most patients with COPD to easily access these medications. However, there may be some cases where antibiotics or systemic corticosteroids prescribed for previous exacerbations are still available and used during the current exacerbation. Currently, in several large clinical trials of COPD medications, the definition of moderate exacerbations often includes worsening symptoms requiring antibiotic and/or systemic corticosteroid treatment without specifically emphasizing the need for outpatient visits (N Engl J Med. 2020;383(1):35-48. N Engl J Med. 2023;389(3):205-214). This differs from the definition used in this study. The initial decision to adopt this definition of exacerbations was primarily to maintain consistency with previous studies like Tie-COPD study. After recognizing this inconsistency, we reviewed every record of moderate exacerbations in our study. We confirmed that patients with moderate exacerbations in this study received antibiotics and/or systemic corticosteroids, and all visited outpatient clinics. Therefore, although the expression may differ from other definitions of moderate exacerbations, the ultimate assessment results remain consistent in this study. We have provided additional clarification on the definition of moderate

exacerbations in the methodology section.

Changes in the text: We have modified our manuscript as advised (see Page 17, lines 374-389 in the revised manuscript).

Comment 3: *In clinical trials, mild exacerbations are often captured using validated daily symptom questionnaires (or diaries) – I presume this was not done; recording daily symptom scores over two years would have been difficult to sustain. However, reliance on self-reported exacerbations in this study may have compromised data quality, particularly for mild events. The methods only refer to a diary card to record symptoms during exacerbations – please clarify and include among limitations as appropriate.*

Reply: This clinical trial was challenging to conduct because patients with mild-to-moderate COPD have lower adherence compared to advanced COPD patients with frequent exacerbations. Patients with mild-to-moderate COPD generally have fewer respiratory symptoms and are less likely to receive medication treatment in the past. Therefore, the sub-centers selected prior to the study were those with abundant clinical trial experience or previous participation in clinical trials of tiotropium for the treatment of mild to moderate COPD (Tie-COPD). These research centers strictly adhered to the study protocol, collecting and reviewing diary cards every three months and issuing new ones. Additionally, researchers at each sub-center provided training every three months to patients on the proper use of the diary cards to ensure the most accurate recording possible. During each follow-up visit, we also inquire about medication usage

and respiratory symptoms in the previous three months. We made every effort to identify mild exacerbations using the available means. However, there may still be underdiagnosed in mild exacerbations. Nevertheless, as this study is a double-blind parallel-group drug clinical trial, the issue of underdiagnosis of mild exacerbations should be equally applicable to both the N-acetylcysteine group and the placebo group. In summary, we made every possible effort to identify mild -exacerbations. Even if mild exacerbations were missed, the risk of missing them would be similar in both the N-acetylcysteine group and the placebo group. The proportion of mild acute exacerbations in all acute exacerbations in this study was the same as in previous studies (Eur Respir J. 2017;50(4):1700711. N Engl J Med. 2017;377(10):923-935). Therefore, the potential limitations in diagnosing mild exacerbations are unlikely to impact the findings of this study significantly. We have added a description of the underdiagnosis of mild exacerbations in the Limitations of the Discussion section.

Changes in the text: We have modified our manuscript as advised (see Pages 13-14, lines 295-304 in the revised manuscript).

Comment 4: *There is no mention of making any statistical adjustment to account for the selection of two co-primary outcomes; this should be clarified either way. Not making an adjustment could be justified if it was considered that the co-primary outcomes would not lead to a change in clinical practice if either one alone was positive in isolation.*

Reply: We sincerely apologize for not conducting multiple adjustments when

calculating the sample size for the selection of two coprimary endpoints in the study design. We chose the largest sample size calculated for the two coprimary endpoints for this study. To make this point clear to the readers, we have provided supplementary explanations in the sample size calculation section. We appreciate your constructive suggestions.

Changes in the text: We have modified our manuscript as advised (see Page 19, lines 432-434 in the revised manuscript).

Comment 5: *Results: consider providing the number need to treat to prevent one moderate or severe, and to prevent one severe exacerbation per year. This will give a better understanding of the treatment burden versus benefit.*

Reply: We additionally performed the number need to treat (NNT) analysis. The corresponding analysis results are presented to the Results section of the manuscript.

Changes in the text: We have modified our manuscript as advised (see Page 9, lines 190-193, and Page 20, lines 442-444 in the revised manuscript).

Comment 6: *The authors highlight that the relatively high dropout rate may reflect the low symptom burden (and infrequent exacerbations) in this population. Adherence to treatment is likely to be further compromised in the general population compared to those consenting to participate in research. This is a further limitation.*

Reply: Patients with mild-to-moderate COPD who have fewer respiratory symptoms experience a rapid decline in lung function and still suffer from a certain proportion of

acute exacerbations. There is an urgent need for targeted pharmacological treatments. It is challenging for patients with mild-to-moderate COPD, who have fewer symptoms, to adhere to medication largely due to the lack of effective drugs that can slow down or halt disease progression. Taking reference from drug trials for asymptomatic hypertension, pharmacological clinical trials should take the lead in clinical treatment. If there are medications available that can delay or halt disease progression, it is believed that with patient education, long-term adherence to medication can be achieved in patients with mild-to-moderate COPD.

Moreover, most clinical trials face inconsistencies with the real world, especially medication compliance. The vast majority of patients included in clinical trials are patients with good compliance and willing to take active treatment. In the real world, the proportion of patients who take medications regularly and are willing to take active treatment is not high. This is indeed a limitation, but it is one that all clinical trial studies face. Therefore, given the word limit of the manuscript, we decided not to include this limitation in the Discussion section. Thanks again for your constructive comments.

Changes in the text: Without modification.

Comment 7: *Unfortunately, no health-economic analysis was included. This limits the ability to assess whether this study should lead to a change in clinical practice.*

Reply: China's healthcare insurance system is relatively complex. Different occupational groups receive different treatment and benefits. Therefore, health-

economic analysis was not considered during the design of this study. We apologize for the inconvenience, and we are currently unable to conduct a supplementary health-economic analysis.

Changes in the text: Without modification.

Comment 8: *Minor comments: - Abstract line 81-83: the term reduction implies statistical significance – I appreciate the authors acknowledge this was not achieved. Consider rewording – e.g. there was a trend towards..... - A statement that there were between group differences should be limited to those achieving statistical significance. It then becomes unnecessary to include “statistically significant” in relation to such results, though a p value can be shown. - Line 122: define FAS when first used – line 122 not 125. - Line 216: delete “with” (or change to “treatment with”).*

Reply: We have made revisions to address all the points mentioned in your suggestions.

Changes in the text: We have modified our manuscript as advised.

Reviewer 3

Comment 1: *Remarks to the Author: Thank you for inviting me to review the manuscript 'Effect of high-dose N-acetylcysteine on exacerbations and lung function in patients with mild-to-moderate chronic obstructive pulmonary disease: a multicentre, double-blind, randomised, placebo-controlled trial', conducted to assess the efficacy of N-acetylcysteine, compared to matched placebo, primarily in reducing the annual rate of exacerbations and change from baseline to 24 months in FEV1. As a statistical reviewer, this review focuses mainly on the study design, data analysed, methodologies used, and presentation of the results. Overall, the study design is appropriate to answer the research questions being studied. However, there are some details related with the statistical analyses that should be clarified further in order to ensure that the statistical methods used for specific outcomes are appropriate and the results valid.*

Reply: Thank you for your kind comments. We respond point-to-point to your comments below.

Changes in the text: Without modification.

Comment 2: *It is unclear for which specific test the sample size calculation for annual exacerbation was performed. Please include enough details to allow replication of the calculation.*

Reply: Thank you for your kind comments and suggestions. We employed Poisson regression analysis for the primary outcome of exacerbation. Thus, we determined the required sample size for comparing the differences between the two groups using

Poisson regression analysis. The specific calculation process and results using the PASS 15.0 software are illustrated in the figure below. We have provided some supplementary explanations in the sample size calculation section to make the process clearer.

Changes in the text: We have modified our manuscript as advised (see Page 19, lines 422-423 in the revised manuscript).

Comment 3: *The sample size calculation for exacerbation rate was performed assuming a difference of at least 0.23 per patient-year in annual exacerbation rate. Although the authors base this value in the results of previous studies, there should also be a stronger clinical rationale behind choosing this difference. Is a 0.23 difference a clinically significant difference between groups that would invite the clinical world to change practice? According to the reported results there is a statistically significant difference of 0.11 in the annual rate of moderate-to-severe exacerbations. Although statistically significant, is it clinically significant as it considerably less than 0.23? As*

this is the most relevant result of the study, it would be- important to add a reflection on whether any of these values are clinically relevant and the impact it would have on clinical practice.

Reply: Currently, there is minimal clinically significant difference analysis for lung function and CAT scores. However, for exacerbations, the minimal clinically significant difference has not been validated yet (Am J Respir Crit Care Med. 2014;189(3):250-255). Clinical trials of drugs for chronic obstructive pulmonary disease (COPD) with exacerbations as the primary endpoint generally analyze efficacy differences based on a reduction of 15-25% in exacerbation rates (ETHOS study [N Engl J Med. 2020;383(1):35-48]: 15% reduction; BOREAS study [N Engl J Med. 2023;389(3):205-214]: 25% reduction; RESTORE study [Eur Respir J. 2017;50(4):1700711]: 20% reduction). In this study, we calculated the sample size using Poisson regression analysis to detect a difference of 0.23 per patient-year between the N-acetylcysteine group (0.27 per patient-year) and the placebo group (0.50 per patient-year) in the annual rate of exacerbations. The reduction of 0.23 per patient-year exacerbations that we set is a 46% decrease from the exacerbation rate in the placebo group. Therefore, considering the overall design of previous clinical trials on COPD drugs, we believe that a reduction of 0.23 per patient-year exacerbations is clinically significant.

Why is there a statistically significant difference in the reduction of 0.11 per patient-year moderate-to-severe exacerbations with N-acetylcysteine compared to the placebo group? This is because our study calculated the sample size considering the differences in coprimary endpoints. When calculating the sample size based on the

inter-group difference in exacerbations, each group required 230 patients, while for the inter-group difference in lung function, each group required 489 patients. In our study, we guided patient enrollment based on the largest calculated sample size. Therefore, the number of enrolled patients in our study exceeded the sample size required for assessing the inter-group difference in exacerbations. Moreover, the reduction of 0.23 per patient-year exacerbations refers to the overall exacerbation rate, whereas the reduction of 0.11 per patient-year exacerbations specifically pertains to moderate-to-severe exacerbations. The reduction of 0.11 per patient-year exacerbations represents a 24% ($0.11 / 0.45$) decrease in moderate-to-severe exacerbations in this study. Considering the differential requirements for reduction in exacerbation rates in previous clinical trials on COPD drugs (15-25%), we believe that a 24% reduction in the occurrence of moderate-to-severe exacerbations holds clinical significance, particularly when considering the cost-effectiveness of N-acetylcysteine as a drug option.

Thank you again for your kind and constructive comments

Changes in the text: Without modification.

Comment 4: *Using only Poisson regression for modelling exacerbation rates raises some concerns as Poisson regression has assumptions, more importantly of equi-dispersion, and the paper provides no evidence that the assumptions were checked. Since inference from a Poisson regression relies upon the assumption of equi-dispersed data, resulting effects tests are biased when the data are truly over- or under-dispersed. If over- or under- dispersion is present, different distributions should be assumed (e.g.*

negative binomial). The authors mention correction for overdispersion, including it to the list of variables adjusted in the model, but it is unclear how this correction was undertaken, and if the data was indeed over-dispersed. Please clarify or re-analyse the data if necessary.

Reply: The statistical analysis of exacerbations in clinical trials is challenging. With clinical trials carried out in COPD, more understanding is gained regarding how exacerbation rates are distributed within a population. Using conventional Poisson regression analysis alone is insufficient for analyzing inter-group differences in exacerbations in patients with COPD. According to previously published statistical analysis recommended for COPD exacerbations, they recommend using a Poisson regression model that corrects for overdispersion to analyze COPD exacerbations (Statistical treatment of exacerbations in therapeutic trials of chronic obstructive pulmonary disease. Am J Respir Crit Care Med. 2006;173(8):842-846). Moreover, considering that several previous large-scale COPD drug trials have utilized Poisson regression models that correct for overdispersion to analyze inter-group differences in exacerbations (N Engl J Med. 2008;359(15):1543-1554. Lancet. 2015;385(9971):857-866. N Engl J Med. 2017;377(10):923-935. Eur Respir J. 2017;50(4):1700711.), we have chosen this statistical analysis approach.

We greatly value your comments. Dr. Wu conducted the analysis of inter-group differences in exacerbations using a negative binomial regression model, and the results were similar to the current analysis results. In order to maintain consistency with the previously specified statistical analysis plan, we have decided to present the results

obtained through the overdispersion-corrected Poisson regression model. If necessary, we can supplement the results of the negative binomial analysis in the appendix.

Thank you again for your kind and constructive comments

Changes in the text: Without modification.

Comment 5: *Regarding the repeated measures analysis of variance (ANOVA), to compare the differences in FEV₁ and FVC before and after bronchodilator use at each visit between the two groups, it can be somewhat misleading to call it as such when, by including an interaction term, this is equivalent to a mixed-effects model (a.k.a. multilevel or hierarchical model). This should be clarified as the readership might be more familiar with the latter terms.*

Reply: We used a mixed-effects model for repeated measures to analyze the differences in lung function across different time points. This analysis method is also referred to as repeated measures analysis of variance (ANOVA). Indeed, it would be more appropriate to use a more commonly recognized term to avoid ambiguity. We have made modifications to the statistical analysis section based on your suggestions.

Changes in the text: We have modified our manuscript as advised (see Page 20, lines 445-446 in the revised manuscript).

Comment 6: *Regarding imputation of missing data, please report the percentage of missing data for each variable included in the models. Imputation is only recommended in certain scenarios and if missing data is less than a specific threshold. Useful*

recommendations can be found in this paper: Jakobsen et al, 2017, BMC Medical Research Methodology, volume 17, 162 (2017).

Reply: No missing values were observed for the variables presented in Table 1, which includes baseline characteristics. For both the efficacy and safety analyses, we have indicated the data availability for each time point. Except for using the maximum likelihood algorithm to estimate missing data in the random coefficient regression model, no imputation was performed for missing data in the remaining analyses.

Changes in the text: We have modified our manuscript as advised (see Page 21, lines 459-461 in the revised manuscript).

Comment 7: *Although the authors mention imputing missing data in lines 436/437, they then say that no data imputation was necessary in line 440. Is this because no data was missing? Please clarify.*

Reply: We apologize for this confusion. We have provided additional descriptions in the statistical analysis section to clarify our approach. We used a random coefficient regression model to compare the differences in the annual decline in lung function. The maximum likelihood algorithm was used in the random coefficient model to impute missing values. Except for using the maximum likelihood algorithm to estimate missing data in the random coefficient regression model, no imputation was performed for missing data in the remaining analyses.

Changes in the text: We have modified our manuscript as advised (see Page 21, lines 459-461 in the revised manuscript).

Comment 8: *The term ‘transfer tables’ (line 438) that were used to compare the change in the mMRC score is unfamiliar. Please clarify what method was used.*

Reply: We apologize for this confusion. We have modified transfer tables to Cochran's and Mantel-Haenszel test.

Changes in the text: We have modified our manuscript as advised (see Page 20, line 457 in the revised manuscript).

Comment 9: *Please describe the pre-specified subgroup analyses in line 441 and the rationale behind the subgroups based on continuous scores.*

Reply: We have modified the sentence to be more direct and precise. *“We performed prespecified subgroup analyses according to the mMRC score (<2 or ≥2), CAT score (<2 or ≥2), and GOLD stage (1 [mild] or 2 [moderate]). We also performed additional exploratory subgroup analyses according to smoking status (never smoker or ever smoker), use of medication for COPD at baseline (Yes or No), and COPD exacerbations in the previous year at baseline (Yes or No).”*

Changes in the text: We have modified our manuscript as advised (see Pages 18-19, lines 412-413 in the revised manuscript).

Comment 10: *In the Results section, when it says in line 140-142 that ‘The peripheral blood N-acetylcysteine concentration in the N acetylcysteine group was significantly higher than in the placebo group (Extended Data Table 5), these results seem to refer to ‘at 24 months’ with no statistical significance at baseline (and the table mentions*

48 months). Please clarify.

Reply: This is a clerical error. We have revised it to 24 months.

Changes in the text: We have modified our manuscript as advised (see Extended Data Table 5 in the revised manuscript).

REVIEWER COMMENTS

Reviewer #1 (Remarks to the Author):

The authors have addressed the concerns raised, within the limitations of the available data, and substantially improved this manuscript.

Line 190-193: the treatment period should be stated in the NNT analysis results.

Line 248-249: I do not think the statement that "Inexpensive N-acetylcysteine is thus an option, especially in low- and middle-income countries" is justified by the results of this study. Firstly, the study was negative for the primary outcomes, secondly, whilst a reduction in moderate or severe exacerbations is reported in secondary outcomes, there was no difference in severe (hospitalised) exacerbations and no health economic analysis has been performed. The number of people with mild or moderate COPD is very large, thus if such a policy was implemented the cost at a population level would be substantial.

Lines 278-288: whilst the strict lockdown in China prevented widespread circulation of Covid-19, this will also have reduced the circulation of other respiratory viruses and consequently exacerbations (for instance, see: doi.org/10.3390/medicina58010066). The effect will be balanced across both arms, but a lower baseline exacerbation rate will have an impact on the exacerbation rate related outcomes.

When two primary outcomes are selected, an adjustment to the significance level should also be considered if the trial would be considered positive if either outcome was achieved in isolation (for instance, using 0.025 rather than 0.05). Perhaps ask the opinion of the statistician who reviewed this paper.

Use of language should be reviewed, for instance: lines 147-148; 189 (subgroups?); 219 (omit then or change to thus); 243-245; 247-248; 300.

Reviewer #2 (Remarks to the Author):

Thank you for addressing my comments. There are just two comments in need of further clarification.

Comment 3: Following the comprehensive explanation provided by the authors, it would be positive to also justify in the paper, perhaps in the Methods, the use a 0.23 difference and why it is clinically significant. Equally, clarify in the Results, how a reduction of 0.11 is not only statistically significant but also clinically relevant as it represent a reduction of XX% and this is clinically significant for those reasons that you stated.

Comment 4: The choice of methodology should always be guided by the properties of the data being analysed which might differ from the properties of the data analysed in other papers. Using the same methodology will only originate comparable data if assumptions are held in both analyses. Otherwise, results will not be valid and therefore should not be compared. It might be the case that the data in other papers was over-dispersed and therefore an adjustment for over-dispersion was necessary. For the current paper, data might be under dispersed, in which case different methodologies should be used. The data might also not be under or over dispersed and no adjustment is necessary. Therefore, the authors should provide further info, such as the initial Poisson regression dispersion parameter, on whether the data was over or under dispersed and

how that guided the analyses. If the solution was to adjust the Poisson regression results, please describe in detail in the Methods what method was used. If negative binomial regression was used instead, then these results should be presented instead. If the data is under dispersed, then appropriate methods should be used. The results presented should be originated from the most appropriate approach .

Reviewer #3 (Remarks to the Author):

The authors have conducted a study of NAC in patients with mild to moderate COPD. Primary endpoints were negative.

Major queries:

- 1) The study should be described as negative. Both the exacerbation data and the FEV1 decline as predefined clearly did not meet the primary endpoint. The abstract is clear in this respect. The first paragraph of the discussion suggests possible positive subgroups, but I would temper this as these were subgroups without statistical adjustment for numerous comparisons.
- 2) The authors need to be cognizant that they have studied 'mild' COPD and not 'early' COPD. The age of the patients studied is outside of what evolving 'guidelines' are suggesting would be the criteria for chronologically early COPD.
- 3) The authors describe the impact of the COVID-19 pandemic and suggest little impact. Can they provide the actual dates of patient enrollment and final follow-up assessments?
- 4) The authors do an admirable job of discussing the potential impact of dropouts. The main concern is not the impact of dropouts in influencing the NAC vs placebo comparisons but the impact on generalizability as there are suggested differences in patient characteristics between the overall group, the completing group, and the drop outs in key features of disease.
- 5) The nature of the patient population remains unusual given restraints on allowed concomitant medications allowed (e.g. inhaled steroids excluded). I realize that this may represent country specific guidelines but in many other countries ICS are used in patients with mild disease, particularly if there is a history of exacerbations. Similarly, GOLD includes ICS in patients with elevated eosinophil count which is not discussed here.

Minor queries:

- 1) Line 100 – probably better stated as 'It is urgent to decrease exacerbation risk and ...'
- 2) Line 124 – likely better stated as '...completed at least one follow-up spirometry...'
- 3) Lines 154-156 – There was no difference between NAC and placebo in all or severe exacerbations. This suggests that the only possible difference was in moderate exacerbations. What is the biological rationale for this?
- 4) Result section – make sure that you are congruent in the reference to the figures. Figures 3 and 4 are not explicitly referred to, e.g.
- 5) Line 186 – as per prior point this seems to be referring to Figure 4.
- 6) Line 186 – Is the group defined by mMRC, CAT and GOLD Stage 1 a composite population or three separate subgroups?
- 7) Lines 190-3 – I realize a reviewer asked for the NNT data but is it appropriate to present this when the co-primary endpoints were negative?
- 8) Lines 198-202 – See major comment #1.
- 9) Line 205 – see major comment #2 above.
- 10) Line 230 – see major comment #2 above.
- 11) Line 300 – Should probably be 'An underdiagnosis...'

The main corrections in the manuscript and the responses to the reviewers' comments are as follows:

Reviewer 1

Comment 1: *The authors have addressed the concerns raised, within the limitations of the available data, and substantially improved this manuscript.*

Reply: Thank you for your kind comments.

Changes in the text: Without modification.

Comment 2: *Line 190-193: the treatment period should be stated in the NNT analysis results.*

Reply: After being reminded by the reviewer 3 and consulting the statistical experts, they believe that the NNT should be calculated when the primary outcome is positive. It is not recommended to calculate NNT if the primary outcome is negative. Therefore, we have deleted the content of the NNT analysis results from the present manuscript.

The final decision on whether to present the NNT analysis results is left to the editors and reviewers. We are very happy to make additions and modifications based on the opinions of the editors and reviewers.

Changes in the text: We have modified our manuscript as advised (see Page 9, line 190, and Page 20, line 453 in the revised manuscript).

Comment 3: *Line 248-249: I do not think the statement that “Inexpensive N-acetylcysteine is thus an option, especially in low- and middle-income countries” is justified by the results of this study. Firstly, the study was negative for the primary outcomes, secondly, whilst a reduction in moderate or severe exacerbations is reported in secondary outcomes, there was no difference in severe (hospitalised) exacerbations and no health economic analysis has been performed. The number of people with mild or moderate COPD is very large, thus if such a policy was implemented the cost at a population level would be substantial.*

Reply: We have removed unreasonable sentences.

Changes in the text: We have modified our manuscript as advised (see Page 11, line 242 in the revised manuscript).

Comment 4: *Lines 278-288: whilst the strict lockdown in China prevented widespread circulation of Covid-19, this will also have reduced the circulation of other respiratory viruses and consequently exacerbations (for instance, see: doi.org/10.3390/medicina58010066). The effect will be balanced across both arms, but a lower baseline exacerbation rate will have an impact on the exacerbation rate related outcomes.*

Reply: We have further added the description of COVID-19 in the manuscript based on your suggestions. Moreover, it also cited a paper that the prevention and treatment measures of COVID-19 will reduce the incidence of acute exacerbation of COPD.

Changes in the text: We have modified our manuscript as advised (see Page 12-13, lines 272-277 in the revised manuscript).

Comment 5: *When two primary outcomes are selected, an adjustment to the significance level should also be considered if the trial would be considered positive if either outcome was achieved in isolation (for instance, using 0.025 rather than 0.05). Perhaps ask the opinion of the statistician who reviewed this paper.*

Reply: Thank you for your kind comments and suggestions. If we recalculate the sample size based on a risk of type 1 error of 0.025 for two primary outcomes, 362 subjects were needed to detect differences in the incidence of exacerbations, and 772 subjects were needed to detect differences in lung function (without considering the 35% dropout rate) respectively. Unfortunately, the number of patients was sufficient for full analysis set (FAS) for exacerbation, but not sufficient for annual decline of lung function. Furthermore, the two primary outcomes of this study were both negative, and setting 0.025 as the intermediate value for the significance test will not significantly change the conclusion of this study. We decided to faithfully present our sample size calculation process consistent with the statistical analysis plan. Meantime, we clearly stated in the sample size calculation section of the manuscript that we set two primary

outcomes to calculate the sample size without correction for multiple comparisons. Of course, we expected the final decision on whether to modify the type I error to 0.025 based on the opinions of the editors and reviewers.

Changes in the text: We have modified our manuscript as advised (see Page 20, lines 442-443 in the revised manuscript).

Comment 6: *Use of language should be reviewed, for instance: lines 147-148; 189 (subgroups?); 219 (omit then or change to thus); 243-245; 247-248; 300.*

Reply: Thank you for your kind comments. We invited native English-speaking researchers to carefully revise this manuscript.

Changes in the text: We have modified our manuscript as advised (see the revised manuscript).

Reviewer 2

Comment 1: *Thank you for addressing my comments. There are just two comments in need of further clarification. Comment 3: Following the comprehensive explanation provided by the authors, it would be positive to also justify in the paper, perhaps in the Methods, the use of a 0.23 difference and why it is clinically significant. Equally, clarify in the Results, how a reduction of 0.11 is not only statistically significant but also clinically relevant as it represent a reduction of XX % and this is clinically significant for those reasons that you stated.*

Reply: Thank you for your kind comments and suggestions. We have revised the Methods, Results, and Discussion sections of the manuscript accordingly, as you suggested.

Changes in the text: We have modified our manuscript as advised (see Page 7, line 153, Page 11, lines 230-233, and Page 19-20, lines 431-437 in the revised manuscript).

Comment 2: *Comment 4: The choice of methodology should always be guided by the properties of the data being analysed which might differ from the properties of the data*

analysed in other papers. Using the same methodology will only originate comparable data if assumptions are held in both analyses. Otherwise, results will not be valid and therefore should not be compared. It might be the case that the data in other papers was over-dispersed and therefore an adjustment for over-dispersion was necessary. For the current paper, data might be under dispersed, in which case different methodologies should be used. The data might also not be under or over dispersed and no adjustment is necessary. Therefore, the authors should provide further info, such as the initial Poisson regression dispersion parameter, on whether the data was over or under dispersed and how that guided the analyses. If the solution was to adjust the Poisson regression results, please describe in detail in the Methods what method was used. If negative binomial regression was used instead, then these results should be presented instead. If the data is under dispersed, then appropriate methods should be used. The results presented should be originated from the most appropriate approach.

Reply: We very much agree with your comments and suggestions. We further detail our rationale and procedure for using Poisson regression correcting for over-dispersed distributions in the statistical analysis section of Methods.

Changes in the text: We have modified our manuscript as advised (see Page 20, lines 451-453 in the revised manuscript).

Reviewer 3

Comment 1: *The authors have conducted a study of NAC in patients with mild to moderate COPD. Primary endpoints were negative. Major queries: 1) The study should be described as negative. Both the exacerbation data and the FEV₁ decline as predefined clearly did not meet the primary endpoint. The abstract is clear in this respect. The first paragraph of the discussion suggests possible positive subgroups, but I would temper this as these were subgroups without statistical adjustment for numerous comparisons.*

Reply: Thank you for your kind comments. We have removed the description of subgroup analyses and secondary study endpoints in the first paragraph of the

Discussion.

Changes in the text: We have modified our manuscript as advised (see Page 9, line 195 in the revised manuscript).

Comment 2: *The authors need to be cognizant that they have studied ‘mild’ COPD and not ‘early’ COPD. The age of the patients studied is outside of what evolving ‘guidelines’ are suggesting would be the criteria for chronologically early COPD.*

Reply: Thank you for your kind comments. We have revised the description of early COPD to mild disease.

Changes in the text: We have modified our manuscript as advised (see Page 10, line 223 in the revised manuscript).

Comment 3: *The authors describe the impact of the COVID-19 pandemic and suggest little impact. Can they provide the actual dates of patient enrollment and final follow-up assessments?*

Reply: We have added actual enrollment and end of last subject follow-up times to the manuscript. The enrollment period for the patients in this study was from September 7, 2017, to August 29, 2019, and the last patient completed follow-up on January 3, 2022.

Changes in the text: We have modified our manuscript as advised (see Page 12, lines 272-273 in the revised manuscript).

Comment 4: *The authors do an admirable job of discussing the potential impact of dropouts. The main concern is not the impact of dropouts in influencing the NAC vs placebo comparisons but the impact on generalizability as there are suggested differences in patient characteristics between the overall group, the completing group, and the drop outs in key features of disease.*

Reply: Referring to previous clinical trials of COPD drugs, the clinical characteristics of patients who withdraw from the trial are generally different from those of patients who complete the trial because patients with mild-to-moderate COPD who have severe disease severity are more likely to drop out of the trial (Zhou Y, et al. Tiotropium in

Early-Stage Chronic Obstructive Pulmonary Disease. N Engl J Med. 2017;377:923-935). We were mainly interested in the differences in clinical characteristics between the intervention group and the placebo group among patients who dropped out to evaluate the impact of withdrawal on the intervention effect. Thank you very much for your good further comments

Changes in the text: Without modification.

Comment 5: *The nature of the patient population remains unusual given restraints on allowed concomitant medications allowed (e.g. inhaled steroids excluded). I realize that this may represent country specific guidelines but in many other countries ICS are used in patients with mild disease, particularly if there is a history of exacerbations. Similarly, GOLD includes ICS in patients with elevated eosinophil count which is not discussed here.*

Reply: Thank you for your kind comments. We initially considered limiting the concomitant use of ICS because previous studies have shown that NAC is more effective in COPD patients who do not use ICS (Decramer M, et al. Effects of N-acetylcysteine on outcomes in chronic obstructive pulmonary disease (Bronchitis Randomized on NAC Cost-Utility Study, BRONCUS): a randomised placebo-controlled trial. Lancet. 2005;365:1552-1560).

GOLD 2024 currently recommends that COPD patients with elevated blood eosinophils or frequent acute exacerbations use inhaled drugs containing ICS regardless of the patient's lung function classification. Some patients with mild-to-moderate COPD are recommended to inhale drugs containing ICS in clinical practice. However, as shown in our baseline characteristics table (Table 1), the patients with mild-to-moderate COPD selected in this trial came from community screening, had few symptoms, and almost never experienced frequent acute exacerbations. Therefore, currently, very few patients with mild-to-moderate COPD actually inhale drugs containing ICS for a long time, just like a clinical trial from the UK for mild-to-moderate COPD, only about 10% of the patients inhaled steroids (Jolly K, et al. Self management of patients with mild COPD in primary care: randomised controlled trial.

BMJ. 2018;361:k2241).

Changes in the text: Without modification.

Comment 6: *Minor queries:*

1) *Line 100 – probably better stated as ‘It is urgent to decreased exacerbation risk and ...’*

2) *Line 124 – likely better stated as ‘...completed at least one follow-up spirometry...’*

4) *Result section – make sure that you are congruent in the reference to the figures. Figures 3 and 4 are not explicitly referred to, e.g.*

5) *Line 186 – as per prior point this seems to be referring to Figure 4.*

8) *Lines 198-202 – See major comment #1.*

9) *Line 205 – see major comment #2 above.*

10) *Line 230 – see major comment #2 above.*

11) *Line 300 – Should probably be ‘An underdiagnosis...’*

Reply: We have made extensive and detailed revisions to the corresponding parts of the manuscript based on your suggestions.

Changes in the text: We have modified our manuscript as advised (see the revised manuscript).

Comment 7: 3) *Lines 154-156 – There was no difference between NAC and placebo in all or severe exacerbations. This suggests that the only possible difference was in moderate exacerbations. What is the biological rationale for this?*

Reply: Thank you for your kind comments. Our study revealed a statistically significant difference in the rate of moderate-to-severe exacerbation between the two groups, but not in the rate of total exacerbations. This may be explained as follows. First, N-acetylcysteine can reduce moderate-to-severe exacerbation, but this is not as obvious as bronchodilators and other mucolytic/antioxidant drugs. N-acetylcysteine may reduce the severity of moderate-to-severe exacerbations, turning them into mild exacerbations, resulting in no significant between-group difference in the rate of total exacerbations. Second, the judgment and reporting of moderate-to-severe exacerbations are clear,

while mild exacerbations may be underreported. For this reason, the most recent COPD clinical trials have used the annual rate of moderate-to-severe exacerbation as the primary study endpoint.

Changes in the text: We have modified our manuscript as advised (see Page 11-12, lines 244-253 in the revised manuscript).

Comment 8: 6) Line 186 – *Is the group defined by mMRC, CAT and GOLD Stage 1 a composite population or three separate subgroups?*

Reply: Subgroup analysis was performed in these three separate subgroups. We have modified the sentences in the manuscript to express them more clearly.

Changes in the text: We have modified our manuscript as advised (see Page 9, line 184 in the revised manuscript).

Comment 9: 7) Lines 190-3 – *I realize a reviewer asked for the NNT data but is it appropriate to present this when the co-primary endpoints were negative?*

Reply: Thank you for your suggestion. We further consulted a number of statistical experts. They agreed that the NNT should be calculated when the primary study endpoint is positive (Monnier L, et al. Number Needed-to-Treat (NNT): Is it a necessary marker of therapeutic efficiency? *Diabetes Metab.* 2020;46:261-264). It is not recommended to calculate NNT if the primary study endpoint is negative.

Changes in the text: We have modified our manuscript as advised (see Page 9, line 190, and Page 20, line 453 in the revised manuscript).

REVIEWERS' COMMENTS

Reviewer #1 (Remarks to the Author):

The authors have addressed the concerns raised effectively.

NNT: as the study was negative for the co-primary outcomes of total exacerbations (including mild events) and FEV1, I agree it is reasonable to omit the NNT analysis for the significant reduction in moderate or severe exacerbations (secondary outcome). In hindsight, moderate or severe exacerbations are more clearly defined and this may have been a better primary outcome. This has also been addressed.

Discussion, line 319 forward: in terms of future trials, it may be worth assessing the efficacy of mucolytics in patients with mild to moderate COPD, chronic bronchitis phenotype and history of exacerbations within the previous 12 months, selecting moderate or severe exacerbations as the sole primary outcome measure.

Minor comments

146: Please rephrase: "There was a tendency for N-acetylcysteine reduced the...".

238 – 240: the authors now correctly highlight the need for caution interpreting secondary outcomes when the (co-)primary outcomes are negative, favouring omitting this statement. If retained, use of language requires revision.

Line 300: "...which may be underdiagnosed".

Reviewer #2 (Remarks to the Author):

The authors have addressed my comments satisfactorily and I therefore have no further comments.

The main corrections in the manuscript and the responses to the reviewers' comments are as follows:

Reviewer 1

Comment 1: *The authors have addressed the concerns raised effectively. NNT: as the study was negative for the co-primary outcomes of total exacerbations (including mild events) and FEV1, I agree it is reasonable to omit the NNT analysis for the significant reduction in moderate or severe exacerbations (secondary outcome). In hindsight, moderate or severe exacerbations are more clearly defined and this may have been a better primary outcome. This has also been addressed.*

Reply: Thank you for your kind comments.

Changes in the text: Without modification.

Comment 2: *Discussion, line 319 forward: in terms of future trials, it may be worth assessing the efficacy of mucolytics in patients with mild to moderate COPD, chronic bronchitis phenotype and history of exacerbations within the previous 12 months, selecting moderate or severe exacerbations as the sole primary outcome measure.*

Reply: Thank you for your kind comments. We appreciate your suggestions and have incorporated them into the revised manuscript.

Changes in the text: We have modified our manuscript as advised (see Page 14-15, line 317-320 in the revised manuscript).

Comment 3: *Minor comments 146: Please rephrase: "There was a tendency for N-acetylcysteine reduced the... "*

Reply: We rephrase this sentence as: The incidence of moderate to severe exacerbations was lower in the N-acetylcysteine group than in the placebo group, but this difference did not reach statistical significance (0.65 vs. 0.72 per patient-year; relative risk [RR], 0.90; 95% confidence interval [CI], 0.80–1.02; P = 0.10) (Fig. 2).

Changes in the text: We have modified our manuscript as advised (see Page 7, line 146-149 in the revised manuscript).

Comment 4: 238 – 240: *the authors now correctly highlight the need for caution interpreting secondary outcomes when the (co-)primary outcomes are negative, favouring omitting this statement. If retained, use of language requires revision.*

Line 300: “...which may be underdiagnosed”.

Reply: We have detected this statement.

Changes in the text: We have modified our manuscript as advised (see Page 10, lines 226 in the revised manuscript).

Comment 5: *Line 300: “...which may be underdiagnosed”.*

Reply: We modified the word.

Changes in the text: We have modified our manuscript as advised (see Page 14, lines 298 in the revised manuscript).

Reviewer 2

Comment 1: *The authors have addressed my comments satisfactorily and I therefore have no further comments.*

Reply: Thank you for your kind comments and suggestions.

Changes in the text: Without modification.